

# Mapping seasonal glacier melt across the Hindu Kush Himalaya with time series synthetic aperture radar (SAR)

**Corey Scher**[1,2], **Nicholas C. Steiner**[2], and **Kyle C. McDonald**[1,2,3]

[1]Department of Earth and Environmental Sciences, The Graduate Center,
City University of New York, New York, NY 10031, USA
[2]Department of Earth and Atmospheric Sciences, The City College of New York,
City University of New York, New York, NY 10031, USA
[3]Carbon Cycle and Ecosystems Group, Jet Propulsion Laboratory, California Institute of Technology,
4800 Oak Grove Drive, Pasadena, CA 91001, USA

**Correspondence:** Nicholas C. Steiner (nsteiner@ccny.cuny.edu)

**Abstract.** Current observational data on Hindu Kush Himalayas (HKH) glaciers are sparse, and characterizations of seasonal melt dynamics are limited. Time series synthetic aperture radar (SAR) imagery enables detection of reach-scale glacier melt characteristics across continents. We analyze $C$-band Sentinel-1 A/B SAR time series data, comprised of 32 741 Sentinel-1 A/B SAR images, and determine the duration of seasonal glacier melting for 105 432 mapped glaciers (83 102 km$^2$ glacierized area, defined using optical observations) in the HKH across the calendar years 2017–2019. Melt onset and duration are recorded at 90 m spatial resolution and 12 d temporal repeat. All glacier areas within the HKH exhibit some degree of melt. Melt signals persist for over half of the year at elevations below 4000 m a.s.l. and for nearly one-quarter of the calendar year at elevations exceeding 7000 m a.s.l. Retrievals of seasonal melting span all elevation ranges of glacierized area in the HKH region, extending greater than 1 km above the maximum elevation of an interpolated 0 °C summer isotherm and at the top of Mount Everest, where in situ data and surface energy balance models indicate that the Khumbu Glacier is melting at surface air temperatures below −10 °C. Sentinel-1 melt retrievals reflect broad-scale trends in glacier mass balance across the region, where the duration of melt retrieved in the western Himalaya and Karakoram is on average 1 month less than in the eastern Himalaya sub-region. Furthermore, percolation zones are apparent from meltwater retention indicated by delayed refreeze. Time series SAR datasets are suitable to support operational monitoring of glacier surface melt and the development and assessment of surface energy balance models of melt-driven ablation across the global cryosphere.

## 1 Introduction

Global warming driven by the anthropogenic release of geologic carbon is causing mass loss of alpine glaciers worldwide (Brangers et al., TS2; Zemp et al., 2006). The Hindu Kush Himalaya (HKH) region, known colloquially as the "Third Pole," has the most ice-covered area on Earth after the high-latitude polar regions (Yao et al., 2012). In contrast to large ice sheets near the poles, these relatively small alpine glaciers – perched at some of the highest elevations on Earth – are among the most sensitive indicators within the global cryosphere of changes in global climate (Anthwal et al., 2006). Just as the recession of these sensitive mountain glaciers contributes to over one-quarter of global sea level rise (Zemp et al., 2019), disturbances accompanying HKH glacier retreat pose innumerable hazards to humans and natural ecosystems. Glacier retreat threatens to disturb the dynamics of river systems delivering freshwater resources to nearly 2 billion people across South and Central Asia (Brown et al., 2007; Milner et al., 2017). Outburst floods resulting from glacier mass loss have killed at least 6300 people in the Himalayas alone and have caused extensive damage to property and livelihoods. These outbursts are expected to in-

crease in frequency with continued glacier wasting (Carrivick and Tweed, 2016). Some organisms endemic to alpine aquatic ecosystems may become extinct as they lose biogeochemical regulation from upstream glaciers (Jacobsen et al., 2012). As global temperatures rise, and perennial snow and ice cover decreases, societies are faced with difficult decisions around the costs and benefits of adapting to a changing climate within and around the HKH region (Brown et al., 2007). Informed decision-making for successful climate change adaptation will require knowledge of the state of natural systems and how these systems are projected to change alongside future increases in population and global average temperature (Bogardi et al., 2012).

Substantial uncertainties exist in projected disturbances associated with a changing climate, environment and hydrologic regime across the greater Himalayas due in part to a lack of observations of in situ hydrology and meteorology at high elevations (Hock et al., 2019; Litt et al., 2019; Marzeion et al., 2020). The magnitude and rate of ablation from surface melting is of particular importance as it drives changes in accumulation-zone snow-properties, such as percolation and densification, that feed back into increased melting (Alexander et al., 2019). Surface melting has also been linked to increased englacial temperatures, resulting in faster ice motion (Miles et al., 2018). Although the general trajectory of changes to the HKH cryosphere is understood (i.e., accelerated glacier mass loss on a decadal scale in the central and eastern Himalaya) (Fujita and Nuimura, 2011), a consensus in projecting changes to HKH hydrology is lacking largely because of missing in situ snow and ice monitoring data across these glaciated river basins (Marzeion et al., 2020). However, construction and maintenance of in situ monitoring station networks are costly and labor-intensive because of the complexity of the high-mountain glaciated terrain. Satellite imaging radar retrieval of alpine glacier melt characteristics has long been proposed as a source of data for hydrologic and glaciologic research (Shi et al., 1994). Understanding of surface melting from observation records will enable advanced climate change projections of glacier wasting that require snow property dynamics describing the retention, refreezing and drainage of liquid water within glacier snow and firn (Pritchard et al., 2020).

Recent findings indicate that shortwave radiation drives melting at elevations where air temperatures are perennially below freezing, such as those on Mount Everest, where temperatures never exceed $-10\,°C$ (Matthews et al., 2019, 2020). These in situ findings indicate the degree to which temperature-indexed melt models are underestimating ablation at these elevations using a $0\,°C$ threshold for glacier melting. Further, studies of glacier wasting in High Mountain Asia have shown variability in patterns and magnitude of glacier wasting across sub-regions of the HKH that would be difficult to capture in numerical models using degree-day assumptions (Brun et al., 2017). An observationally based dataset providing characteristics of the glacier surface energy balance is necessary to capture seasonal and regional variability in glacier wasting across the HKH during melt–freeze cycles.

## Snowmelt detection and radar imaging TS3

This study builds on extensive research on microwave scattering from dry and wet snow and associated techniques for snowmelt retrieval from imaging radar data to present an operational monitoring system for spatially resolved glacier surface melt characteristics using synthetic aperture radar (SAR) time series and outlines of glacier area derived from satellite optical imagery across the HKH. Microwave remote sensing has been used to reliably monitor melt patterns across glaciers and ice sheets (Abdalati and Steffen, 2001; Ashcraft and Long, 2007; Jezek et al., 1994b; Steiner and Tedesco, 2014). Because imaging radar is independent of solar illumination and largely unaffected by cloud cover and atmospheric conditions, the fidelity of radar observations is defined by the frequency of the satellite platform's observational opportunities and by the characteristics of the imaging sensor. At $C$-band frequencies, frozen glacier percolation areas are recognized as one of the brightest radar targets on Earth, and glacier surfaces are unambiguous targets for determination of surface melt/freeze CE1 characteristics (Jezek et al., 1994; Rott and Mätzler, 1987). Detection of seasonal melt on ice surfaces at $C$-band frequencies (4–8 GHz) depends on a strong radiometric response at melt onset (MO), when liquid water content introduced to an otherwise frozen snow or firn matrix causes a drastic decrease in the radar backscatter from the medium (Hallikainen et al., 1986). Deep, frozen snow and firn have a high scattering albedo across microwave frequencies (Matzler, 1998), resulting in high radar backscatter intensity over glaciated regions during the frozen months (Winsvold et al., 2018; Wiscombe and Warren, 1980). The introduction of liquid water in the snow or firn matrix at even hydrologically minimal amounts causes a pronounced increase in the medium's dielectric constant, increasing radar signal attenuation and diminishing volume scattering and leading to a pronounced decrease in radar backscatter, usually by half power or more (Kendra et al., 1998; Shi and Dozier, 1995). In areas that are seasonally snow-free, e.g., for glacier ablation areas of debris cover or bare ice, melting conditions are dominated by heterogeneous scattering mechanisms following the disappearance of seasonal snow, a topic of study not well represented in the theoretical literature on radar physics, likely due to the complex nature of the glacier ablation surface. Because of the relatively strong signal produced at the onset of melting, radar-based melt detection records have been developed across regions of the global cryosphere for several decades using both real and synthetic aperture radar sensors (Bhattacharya et al., 2009; Bindschadler et al., 1987; Koskinen et al., 1997). Subsequently, snowmelt detection algorithms have been developed using a host of radar sensors to monitor the

onset and duration of snowmelt across glaciers and ice sheets (Abdalati and Steffen, 2001; Ashcraft and Long, 2007; Bahr et al., 1997; Jezek et al., 1994; Kayastha et al., 2019; Koskinen et al., 1997; Winebrenner et al., 1994). Prior applications of SAR mapping of seasonal surface melting over ice sheets and glaciers have been limited by a lack of repeat observations such as those now available from the Sentinel-1 SAR constellation (Lund et al., 2019).

Observations from time series SAR data have been used to delineate zones of glacier facies and regions of glacier mass balance (Winsvold et al., 2018). In glacier percolation zones, seasonally wet snow refreezes into ice lenses, pipes, and other percolation-related features that amplify both surface and volume scattering of $C$-band radar and result in the brightest SAR backscatter being captured during the frozen periods (Jezek et al., 1994; Rau et al., 2000). Studies have shown that SAR backscatter generally increases with elevation across glacier surfaces during frozen periods, from the glacier terminus, through zones of ablation and frozen meltwater percolation, and eventually attenuating in zones where dry snow accumulates (Winsvold et al., 2018). In transitions between these zones there are pronounced backscatter contrasts rather than smooth, gradual transitions. At $C$-band frequencies, radar scattering within glacier percolation areas dominates the backscatter amplitude during frozen periods (Jezek et al., 1994). Importantly for melt retrievals, the diminished volume scattering during surface melting in areas of meltwater percolation creates a pronounced and unambiguous radar signature in time series observations. The sensitivity of SAR backscatter to the introduction of liquid water in an otherwise frozen snowpack or firn structure provides a reliable mechanism for the retrieval of percolation zone melt characteristics (Lievens et al., 2019). In refreezing percolation zones, the upper layers of firn will freeze first, with the freezing front advancing downward across layers, thus progressively increasing backscatter and with decreasing total-column liquid water content (Ashcraft and Long, 2005). In this way, the timing of refreeze relative to the surface energy balance at the surface provides a direct and spatially resolved indicator of subsurface meltwater storage within the snow or firn and delineates the percolation zones over mountain glaciers. Like in the accumulation zone, the surface melting response in the ablation zone will dominate the seasonal trends in backscatter because of absorption from liquid water at the surface over both bare-ice and debris-covered portions of ablation areas. Although the absolute fraction of backscatter at $C$-band frequencies over debris-covered portions of ablation zones attributed to volume scatter is not well known, there is evidence that for low frequencies it can account for a majority of radar observations (Huang et al., 2017 TS4).

This study enlists SAR data acquired at a spatiotemporal resolution that captures melt variability across mountain glacier surfaces suitable for constraining seasonal characteristics of melt onset and duration while building on associated methods often employed for glaciers and ice sheets.

In this paper we utilize SAR data to retrieve melt status on HKH glacier surfaces with a simple threshold-based change detection classification melt/freeze state CE2 – an observational constraint on glacier ablation. It is possible that intense incident solar radiation is driving these melt processes at elevations above the 0 °C summer isotherm (Matthews et al., 2019) across the entirety of the HKH and that the sensitivity of SAR backscatter to changes in the glacier surface melt/freeze condition as seen when water transitions between solid and liquid phases provides a real alternative to temperature–elevation lapse rate estimates of melting (Litt et al., 2019) for assessing models of glacier ablation. Though coarse in temporal resolution relative to typical meteorological datasets, retrieval of melt status using SAR time series produces mappings with very high spatial resolution and a continuous record of melt timing and duration across glaciated regions. We present an application of this melt retrieval technique at the scale of the HKH with spatiotemporal fidelity adequate for capturing seasonal variability in melt timing and duration across individual glacier surfaces and sub-regional heterogeneities across the HKH.

## 2 Setting and data

The HKH region (Fig. 1) spans $13 \times 10^6$ km$^2$ CE3, including areas inhabited by 240 million people, with nearly 2 billion people relying on the delivery of water resources from catchments that originate within the region (Scott et al., 2019). Within the high-elevation HKH, seasonal meltwater from snow and glacier ice is the primary source of domestic freshwater supply (Bolch et al., 2012). Wasting of HKH glaciers poses a risk to the domestic water resource supply for those populations living within these high-elevation HKH catchments (Wood et al., 2020). Glacier wasting in the HKH is heterogeneous, and the increase in global average temperature has caused mass wasting of mountain glaciers across all HKH sub-regions (Farinotti et al., 2020; Gardelle et al., 2012). Distinct glacio-climatic sub-regions are characterized by these unique dynamics of glacier wasting (Bolch et al., 2019a). The wasting of HKH glaciers is thus a spatially and temporally heterogeneous phenomenon where distinct glacio-climatic regimes control ablation (Bolch et al., 2012). In this study, we refer to glacio-climate sub-regions delineated in Bolch et al. (2019a) and modified by Shean et al. (2020). These delineations of glacio-climate were produced by the Hindu Kush Himalaya Monitoring and Assessment Program (HiMAP), and we refer to the sub-regional delineations as "HiMAP regions" throughout the text. We selected 12 HiMAP sub-regions that intersected with a boundary of the HKH region delineated by the International Centre for Integrated Mountain Development (ICIMOD). The HKH region, HiMAP sub-regions and glaciated area summaries within each HiMAP sub-region are illustrated in Fig. 1 alongside the Sentinel-1 acquisition plan.

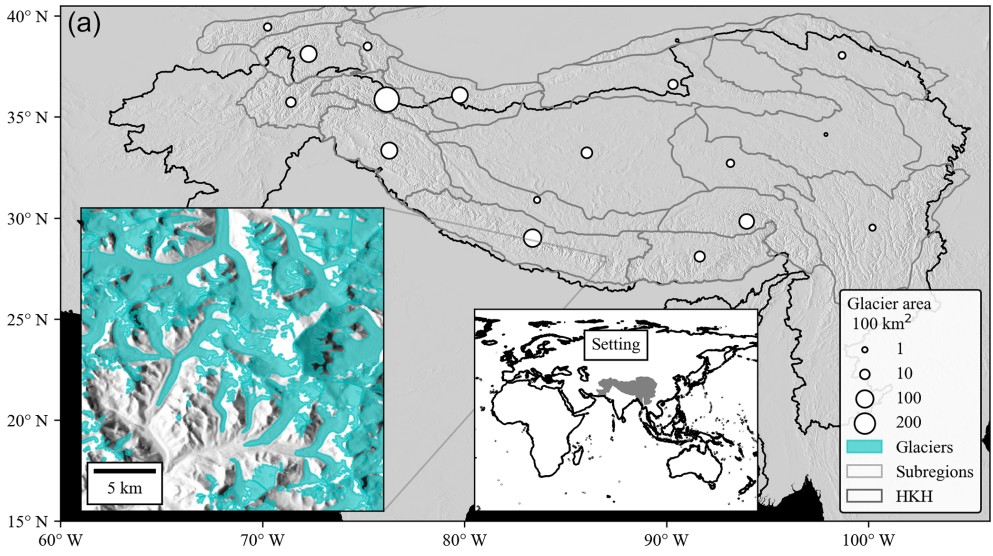

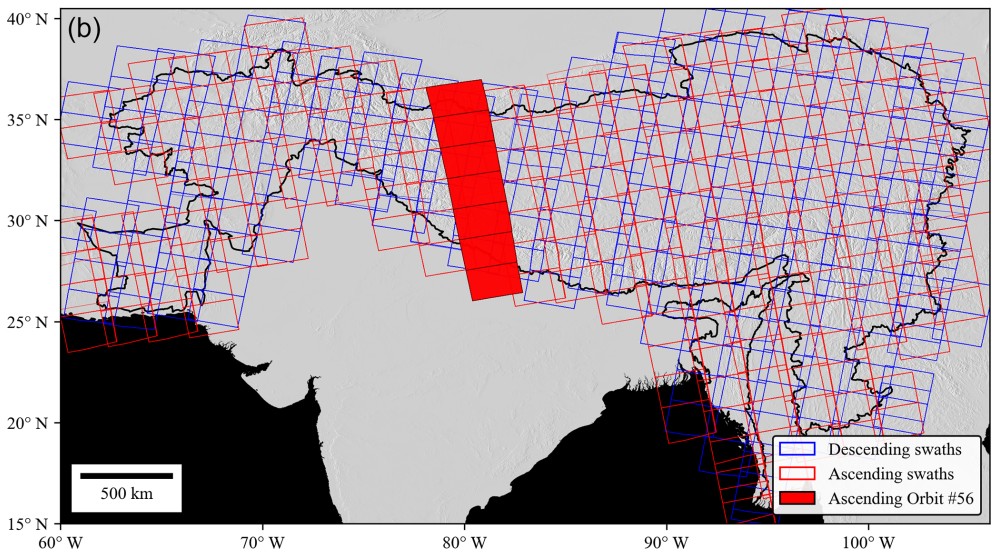

**Figure 1. (a)** Hindu Kush Himalaya (HKH) region and 2018 GAMDAM glacierized areas summed across glacio-climate sub-regions from Shean et al. (2020). An inset map highlights the spatial fidelity of GAMDAM outlines in the top panel. GGI and HKH data overlay a 30 m Shuttle Radar Topography Mission (SRTM) digital elevation model (DEM) hillshade (Farr, 2007). **(b)** Sentinel-1 ascending- (red) and descending-swath (blue) footprints acquired across the study region. Ascending-orbit cycle number 56 is highlighted in red to illustrate the SAR image processing approach for time series analysis across distinct orbit cycles.

## 2.1 GAMDAM glacier inventory (GGI)

The Glacier Area Mapping for Discharge from the Asian Mountains (GAMDAM) glacier inventory (GGI) is a contemporary (July 2019) database on glacier outlines for the region of High Mountain Asia (Fig. 1). These outlines were originally delineated automatically using cloud- and snow-free satellite optical imagery in an initial release of the database (Nuimura et al., 2015). As a recent update to the database, each outline was individually inspected for quality control to correct discrepancies where automatic glacier delineation lost accuracy in terrain-occluded areas, at debris-covered portions of glaciers and through obstruction under seasonal snowpack. The recently updated glacier outlines were derived from satellite optical imagery captured across the HKH by Landsat 5 and 7 between 1990–2010 (Sakai, 2019). Although these data are the most current in terms of quality control spanning the study region, they do not necessarily capture debris-covered portions of glaciers due to confusion with land in optical image classification schemes, an issue that may be resolved with interferometric SAR phase decorrelation (Bolch et al., 2019b). The 2018 GAMDAM

database contained within the HiMAP sub-regions includes 105 432 distinct glacier outlines, spanning a total area of 83 102 km$^2$ within the HKH (Nuimura et al., 2015).

## 2.2 Sentinel-1 synthetic aperture radar

The Sentinel-1 A/B satellites were launched in April of 2014 and 2016, respectively, and collect $C$-band (5.405 GHz) SAR data with a combined revisit interval of 6 d over of the majority of the terrestrial Earth. Each Sentinel-1 scene acquired in the interferometric wide-swath (IW) mode has a width of 250 km and a resolution of $5 \times 20$ m in range and azimuth at the Equator. This study utilized images taken in the IW mode and in a cross-polarized state (VH). Sentinel-1 data were accessed through a cloud-computing platform (discussed below), wherein SAR scenes were radiometrically terrain-corrected to sigma naught backscatter coefficients in decibels (dB) using the European Space Agency's (ESA) Sentinel Application Platform (SNAP) toolbox and the Shuttle Radar Topography Mission (SRTM) 30 m digital elevation model (DEM) (Farr, 2007) upon ingestion into the cloud environment. Data from both the ascending- and descending-orbit nodes were analyzed across the study region for a total consideration of 32 741 individual Sentinel-1 A/B IW scenes across 46 unique orbit cycles captured across the calendar years 2017–2019 (Table 1, Fig. 1b). By combining orbit directions, we utilize observations acquired at day and night. For the purpose of this study we do not attempt to resolve diurnal-scale melt–freeze processes and instead focus on retrieving seasonal and annual characteristics of melt timing and duration. Cross-polarized SAR backscatter provides enhanced observational sensitivity to volume scattering of the radar signal in deep, dense and weathered snowpack and firn (Rott and Mätzler, 1987). We selected cross-polarized (VH) Sentinel-1 A/B observations because VH data show less angular sensitivity to contrasts between dry and wet snow (Nagler et al., 2016). Cross-polarized Sentinel-1 SAR did not become available over the HKH until early 2017 and thus restricted the timeframe of this study. As illustrated in Fig. 2, we observe a large (>3 dB) difference in the seasonal radar backscatter between frozen and melting periods across most glacier surfaces in cross-polarized (VH) SAR data.

## 2.3 Computing infrastructure

A cloud-computing platform and application programming interface (Google Earth Engine) with pre-processed radiometrically terrain-corrected Sentinel-1 A/B data were used to detect melt characteristics across the region (Gorelick et al., 2017). Radiometric terrain correction of Sentinel-1 data was conducted upon ingestion to the cloud server using the ESA's method contained within the Sentinel Applications Platform (SNAP) processing toolbox. The SNAP toolbox is used for Sentinel-1 images to update orbit metadata with restituted orbit files, remove invalid edge data and low-intensity noise, re-

move thermal noise, compute $\sigma^0$ backscatter, and conduct orthorectification upon ingestion of data to the server (Google, 2020). The SNAP toolbox terrain correction functionality utilizes the 30 m spatial resolution SRTM DEM (Farr, 2007; Margulis et al., 2019). The pre-processed SAR time series data and API CE4 functionality used to derive glacier melting characteristics are available from Google Earth Engine and can be used to recreate the work presented in this study.

## 2.4 Automated weather station data

Measurements from two automated weather stations (AWSs CE5) are used to estimate surface energy balance (SEB) and evaluate surface melting conditions over the Khumbu Glacier, and measurements from two additional AWSs are used to calculate temperature–elevation lapse rates for comparison with melt retrievals (Table 2). The Camp and the South Col AWSs were installed around Mount Everest, Nepal, as part of the National Geographic and Rolex Perpetual Planet Expedition to Mt. Everest in April–May 2019 (Matthews et al., 2019). Measurements were collected at an hourly interval and include air temperature, wind speed, relative humidity, incoming shortwave and longwave radiation, and barometric pressure. Time series plots of meteorological observations are shown in Supplement Fig. S1. Please see Matthews et al. (2020) for a complete description of sensor specifications and sampling interval. AWS data collected within the Langtang valley are used to estimate temperature–elevation lapse rates following methods from prior studies and serve as data for comparison with Sentinel-1 backscatter values (Shea, 2016).

## 3 Methods TS5

### 3.1 Classification

We use a threshold-based change detection algorithm applied to time series radar backscatter intensity to classify melt conditions (Ashcraft and Long, 2007). Melt detection is conducted across Sentinel-1 A/B ascending- and descending-orbit track time series separately and mosaicked into a final image based on a statistical score for seasonal melt magnitude after classification. To classify snowmelt, we conduct a pixel-based temporal classification by comparing each image at interval "i" to a dry/frozen CE6 winter average backscatter value calculated from January–February for each study year. Due to missing VH acquisitions at some locations during the 2017 frozen months, (January–February), we utilized 2018 frozen month reference data for melt retrieval across the calendar year 2017 as regular acquisitions across the HKH began in late February 2017. Snowmelt at each image acquisition interval ($m_i$) was classified using Eq. (1):

$$m_i = \begin{cases} 1, & \text{if } \sigma_i^0 < \overline{\sigma}_w^0 - b, \\ 0, & \text{if } \sigma_i^0 > \overline{\sigma}_w^0 - b, \end{cases} \tag{1}$$

Please note the remarks at the end of the manuscript.

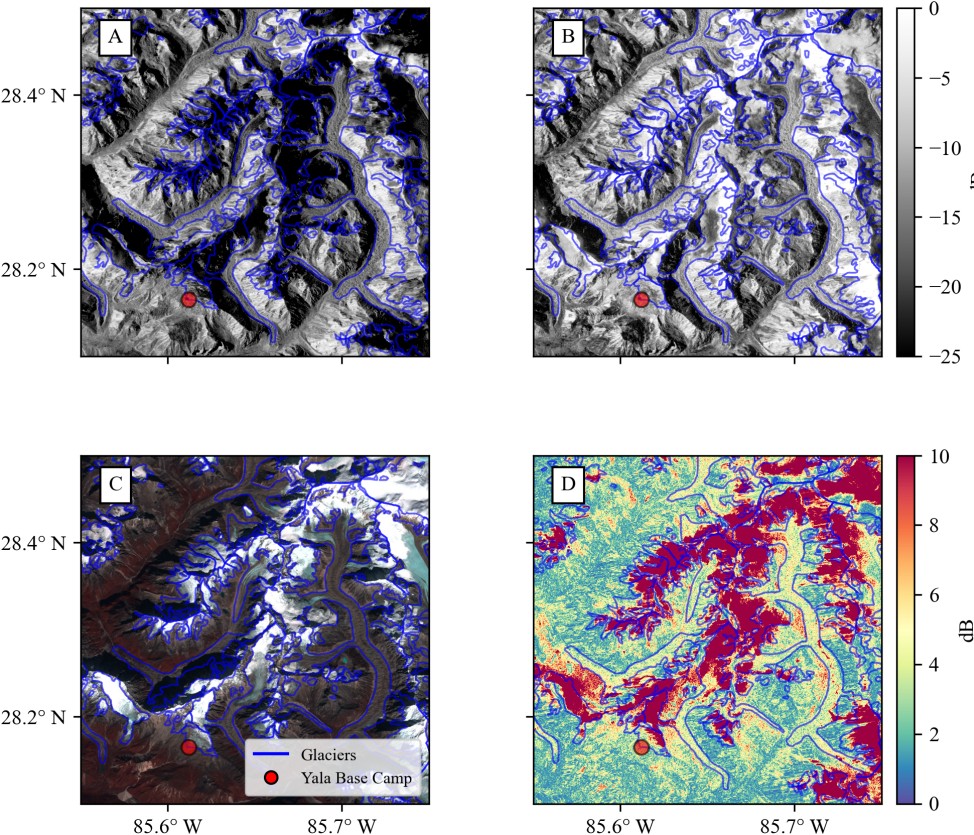

**Figure 2. (a)** Mean summer (July–August) 2018 cross-polarized (VH) backscatter across an example region in the Trishuli basin, Nepal. **(b)** Mean 2018 winter (January–February) VH backscatter from Sentinel-1. **(c)** Sentinel-2 false-color (near-infrared, green, blue) image acquired by Sentinel-2 on 30 October 2018. Glacier outlines are shown in blue, and the Yala Glacier base camp meteorological station is marked in red. Note the snow-covered and bare-ice portions of outlined glaciers and other debris-covered portions of glacier ablation areas. **(d)** The difference between mean summer and winter VH backscatter from Sentinel-1.

**Table 1.** Sentinel-1 image count and orbit paths used in this study.

| Orbit direction | Number of S-1 images by year | | | Relative orbit cycle |
|---|---|---|---|---|
| | 2017 | 2018 | 2019 | |
| Descending | 4424 | 5436 | 5253 | 4, 5, 19, 20, 33, 34, 48,49, 62, 63, 77, 78, 92, 106, 107, 121, 122, 135, 136, 150, 151, 164, 165 |
| Ascending | 5302 | 6097 | 6150 | 12, 13, 26, 27, 41, 42, 55, 56, 70, 71, 85, 86, 99, 100, 114, 115, 128, 129, 143, 144, 158, 172, 173 |

where the ground-range detected backscatter intensity at each image acquisition ($\sigma_i^0$) within the times series must be less than the difference between the mean winter backscatter ($\overline{\sigma}_w^0$) and a fixed threshold ($b$). Threshold values ($b$) have been developed across numerous studies of melt detection with $C$-band scatterometer and SAR datasets using both ground-based observations and radar scattering model results of changes to backscatter magnitude at the onset of melt. We followed previous studies (Baghdadi et al., 1997;

Bhattacharya et al., 2009; Engeset et al., 2002; Nagler and Rott, 2000; Oza et al., 2011; Rott and Mätzler, 1987; Steiner and Tedesco, 2014; Trusel et al., 2012) and selected a $b$ value equal to one-half of the signal power (3 dB). Figure 3 provides an illustration of the SAR melt signal for a high-elevation (4950 m a.s.l.) meteorological station, located at the Yala Glacier base camp. Backscatter values averaged across the Yala Glacier acquired along the Sentinel-1 A/B descending-orbit direction are plotted alongside mean daily

**Table 2.** Sources of air temperature data used to calculate 3 d average temperature–elevation lapse rates within the central Himalaya for the 2018 calendar year.

| Station name | Date range (dd/mm/yyyy) | Resolution | Elevation (m a.s.l.) | Latitude | Longitude | Source |
|---|---|---|---|---|---|---|
| Yala Glacier | 05/08/2012– 12/31/2018 | Hourly | 4950 | 28.23252 | 85.61208 | ICIMOD |
| Kyanging station | 03/22/2012– 12/31/2019 | Hourly | 3,802 | 28.21081 | 85.56169 | ICIMOD |
| Camp II | 05/22/2019– 10/31/2019 | Hourly | 6,464 | 27.9810 | 86.9023 | (Matthews et al., 2019) |
| South Col | 05/22/2019– 10/31/2019 | Hourly | 7,945 | 27.9719 | 86.9295 | (Matthews et al., 2019) |

air temperature recorded at the Yala Glacier base camp automatic weather station (Shea, 2016). If we consider air temperature above 0 °C to control glacier surface melt at this location, classification accuracy for melt retrieval using Eq. (1) is 96 % in the VH polarization.

### 3.2 Quantifying algorithm performance

Sentinel-1 SAR viewing geometry will vary as the local incidence angle increases with across-track range. At high incidence angles (far range), the sensitivity to volume scatter is diminished, and the melting signal is reduced. At $C$-band frequencies, these effects on volume scatter are strongest only at very high incidence angles (closer to grazing) (Nagler and Rott, 2000). We classified areas as valid for melt detection using a metric of statistical separability for seasonal backscatter intensity across frozen and melt periods, which we interpret as a measure of the strength of the seasonal melt signal Eq. (2):

$$z = \frac{\overline{\sigma}_w^0 - \sigma_s^0}{s(\sigma_w^0)},$$ (2)

where the score for seasonal separability of backscatter intensity ($z$) was calculated across each SAR pixel's time series using the difference between the mean winter $\overline{\sigma}_w^0$ (January–February) and summer $\overline{\sigma}_s^0$ (July–August) season backscatter intensities as compared to the standard deviation of backscatter across the winter months ($\sigma_w^0$). In computing $z$, we employed consistent repeat-pass observation geometries, thereby allowing application of the time series melt detection algorithm in regions of complex terrain. This metric serves as a measure of the magnitude of the seasonal melt signal across each pixel's time series. It is used here as a criterion to identify valid melt observations and for selection of pixels employed in regions of overlapping orbital tracks based on the sensitivity of the radar backscatter to melting. We apply this metric to choose which orbit direction (ascending or descending) to use for melt classification on a per-pixel

basis after applying Eq. (1) across each orbit cycle time series so as to capture the maximum area of melt signals occurring across the complex terrain.

Sentinel-1 A/B interferometric wide (IW) swath images have a range in viewing angle between 29.1–46.0° (ESA). Glacier melt retrieval using SAR data commonly begins with a normalization of radar images by viewing angle on a scene-by-scene basis (Adam et al., 1997; Huang et al., 2011; Rott and Mätzler, 1987; Winsvold et al., 2018). We consider changes for each individual orthorectified $10 \times 10$ m pixel time series across distinct, repeating orbit tracks and directions. This approach holds the local incidence angle effectively constant for each region observed by a given set of orbit tracks. Glacier melt classification and $z$-score calculation are carried out across images acquired along identical orbit tracks in distinct orbit directions (Fig. 1) and mosaicked into a final dataset for each study year using the greatest $z$ score observed across each orbit cycle path and in each orbit direction. We thus limit temporal resolution of melt retrievals to 12 d by choosing only observations from the orbit direction with the greater $z$ score on a per-pixel basis. Time series analysis of SAR acquisitions on distinct orbit tracks eliminates the need to normalize each scene by incidence angle for the purposes of melt retrieval. This method reduces computational cost and eliminates artifacts that may originate from overlapping orbit paths and differences in radar viewing angle. Areas where complex topography controls the backscatter should show little time series variability in backscatter change at the SAR pixel scale when viewed at a distinct and consistent orbit path and direction and should not pass the $z$-score test.

We apply time series melt detection only where interseasonal backscatter intensities are separated by greater than 2 standard deviations ($z > 2$), representing better than 98 % confidence in the presence of an annual melt signal. For all locations, the orbit direction and orbit cycle that has the greatest $z$ value is used for melt classification. We find that $z$ generally increases with elevation across sub-regions of the

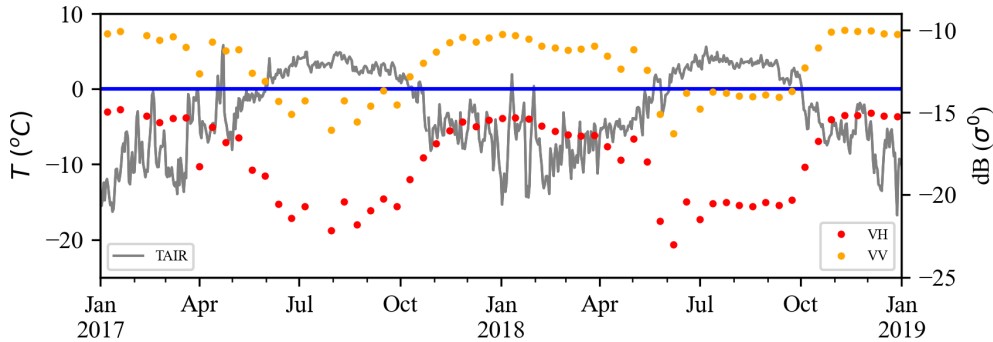

**Figure 3.** Time series chart of air temperature measured at the Yala Glacier base camp (4950 m a.s.l.) and Sentinel-1 A/B descending backscatter averaged across the Yala Glacier for the years 2017–2018. Assessment of algorithm performance assuming mean daily air temperatures above 0 °C indicates active melt results in 96 % accuracy for melt classification across this time series in the VH polarized backscatter.

HKH and that, across elevation ranges, the mean $z$ is above the threshold for melt retrieval, indicating detection of a seasonal melt signal across all ranges of glacier elevation spanning the HKH (Fig. 4). Areas of debris cover may exhibit radar brightening with snow-free conditions above winter mean ($z < 0$). These areas occur towards lower elevations, where seasonal snow or firn does not have a significant contribution to the seasonal backscatter response, and are not included in our melt classification approach following $z$-score thresholding. Nonetheless, there exists retrievable melt signals (i.e., $z > 2$) across ablation surfaces such that median window filtering across ablation zones can result in a geospatial dataset with more complete coverage. We obtain more robust estimates of melting onset and refreeze by spatially aggregating results of the glacier surface melt timing (Eq. 1) using a median window filter of $9 \times 9$ pixels after melt classification and $z$-score validation. Reach-scale regions where SAR signals fail the $z$-score test are thus interpolated over using $9 \times 9$ pixel median window filtering. The complexity of SAR signals involves the diverse scattering mechanisms on ablation surfaces following the disappearance of seasonal snow. Because sufficient data are retrievable on ablation surfaces (i.e., $z > 2$), median window filtering enables greater spatial continuity in SAR-derived melt retrieval data. All spatiotemporal characteristics we report herein are after median window filtering of melt retrievals from 10 m native resolution to 90 m resolution. In Fig. 4 we show the mean $z$ across HiMAP sub-regions in order to illustrate that, on average across HiMAP sub-regions CE7 . Mean seasonal melt magnitude averaged over 100 m elevation bins over all 3 calendar years of data shows strong ($z > 2$) melt signals across glacio-climatic sub-regions and across all elevation ranges of significant glaciation except below $\sim 3400$ m a.s.l. in the eastern Tibetan Mountains CE8 and eastern Hindu Kush sub-regions.

### 3.3 Surface energy balance and surface melting

Sentinel-1 SAR (S1-SAR) detects a substantial area and duration of melting at elevations where air temperatures should be well below freezing. Although measurement data in these areas are scarce, AWSs installed during 2019 at Mt. Everest, Nepal, can provide two instances of point-scale validations of glacier melting using surface energy balance (SEB) modeling based on in situ measurements. As described in Matthews et al. (2020), the highest AWSs on the Earth are installed adjacent to the Khumbu Glacier, Nepal. We use AWS observations to compute SEB described in Matthews et al. (2020). In our SEB modeling, turbulent fluxes are determined using the aerodynamic roughness at the glacier surface taken from measurements in low latitudes (Brock et al., 2006) and evaluated over the 5th to 95th percentile of this sample to capture uncertainty. Surface melting is defined by the glacier surface temperatures ($T_s$) that are evolved from air temperatures and the residual downward glacier heat flux in the iterative approach from TS6 Wheler and Flowers (2011). Melting days are defined where $T_s = 0$ °C at any point during the day. The Supplement for this paper is provided to describe the SEB methodology in further detail (Supplement Sect. S1.1).

A comparison of S1-SAR- and SEB-derived melting is shown in Fig. 5. During 2019, the average daily air temperature measurements at the Camp II station (Fig. 5a) are never above zero but experience above-zero maximum glacier surface temperatures starting in June 2019 and ending in September 2019. At the South Col AWS, the average temperature is much less, close to $-10$ °C on average during summer months (Fig. 5b). S1-SAR estimates of surface melting use two aggregated backscatter time series over $90\,\mathrm{m} \times 90\,\mathrm{m}$ areas where area centers are located nearest to each of the AWS stations over the Khumbu Glacier, Nepal. For the Camp II AWS, this is centered at 6483 m a.s.l. and for the South Col AWS at 7128 m a.s.l. Melting signals are apparent at both Camp II (Fig. 5c) and South Col (Fig. 5d).

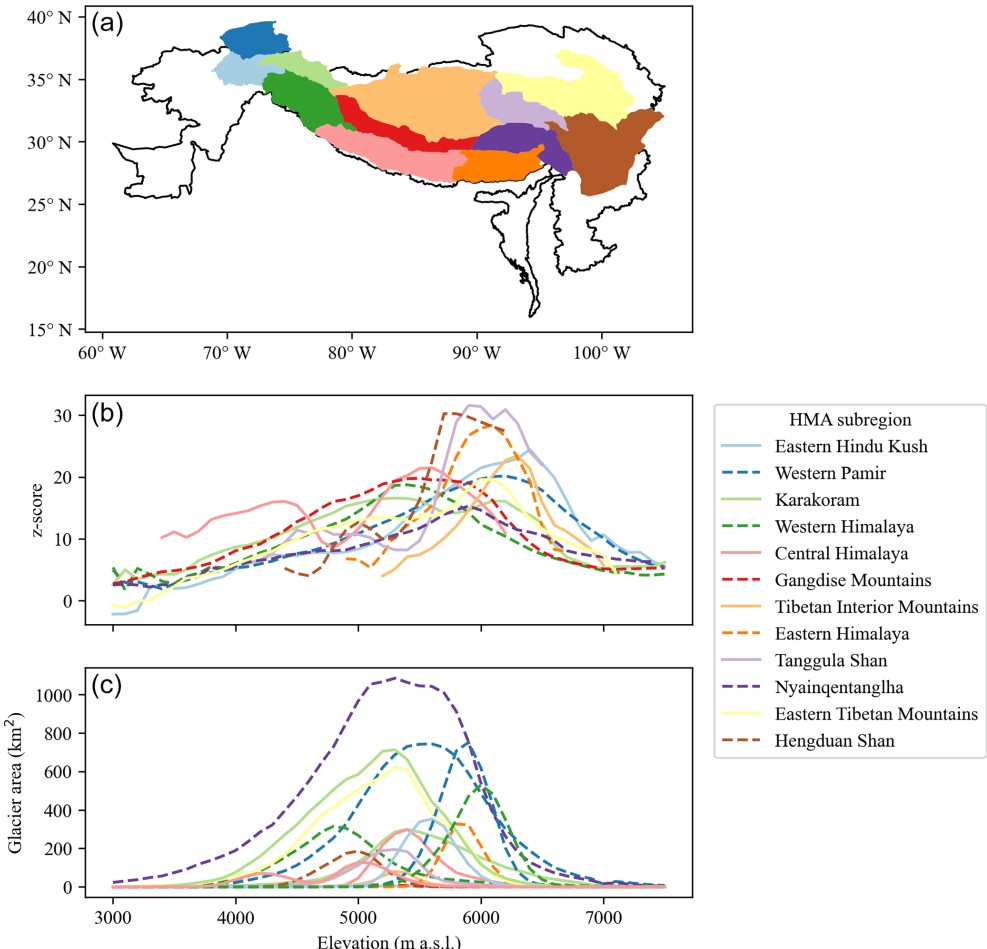

**Figure 4. (a)** Glacio-climate sub-regions within the Hindu Kush Himalaya codified in Shean et al. (2020). **(b)** Mean $z$ score (2017–2019) by 100 m SRTM elevation bin over each sub-region in the HKH. **(c)** Mapped glacier area from the GAMDAM database (Sakai, 2019) over 100 m elevation bins derived using the 30 m SRTM DEM (Farr, 2007) for each sub-region.

Melting is detected at high elevations in both SAR observations and SEB modeling output where daily average air temperatures remain below zero. We find that S1 and SEB estimates of surface melting at the Everest Camp II AWS (6464 m a.s.l.) have an agreement score – the percentage of days where the SEB and SAR find the same condition – that ranges from 73 % to 85 % depending on the parameterization of surface roughness used in SEB estimates of melting. At Mt. Everest South Col (7945 m a.s.l.) the agreement score varies from 63 % to 68 %. We find that the S1-SAR finds 133 d of melting at Camp II, while the SEB indicates from 93 to 100 d. At Mt. Everest South Col the S1-SAR finds 72 d of melting, while the SEB indicates 43 to 56. The start of surface melting at Camp II from SEB modeling is day of year (DOY) 153 and DOY 142 from S1-SAR; at South Col melt onset is DOY 152 from SEB and DOY 146 from S1-SAR. The end of surface melting at Camp II from SEB modeling is DOY 270 and DOY 290 from S1-SAR. At South Col, re-

freeze at the surface from SEB is DOY 256 and DOY 244 from S1-SAR.

Using SEB outputs we find good agreement on surface melt timings; S1-SAR detects melt onset to within 9 d on average at two locations on the Khumbu Glacier in Nepal and refreeze to within 16 d. Although limited by observational data, the agreement in melt duration between S1-SAR and SEB modeling, and the understanding of the physical basis of SAR measurements, we have a high degree of confidence in our methodology and in the ability of the SAR backscatter to detect melting, even CE9 in data-poor regions such as the HKH.

## 3.4 Comparison to temperature–elevation lapse rates TS7

Melting on glacier surfaces across the HKH is controlled by the SEB between the atmosphere and underlaying snow, firn or ice. We explore the relationship between the S1-SAR-derived surface melting record and air temperature–elevation

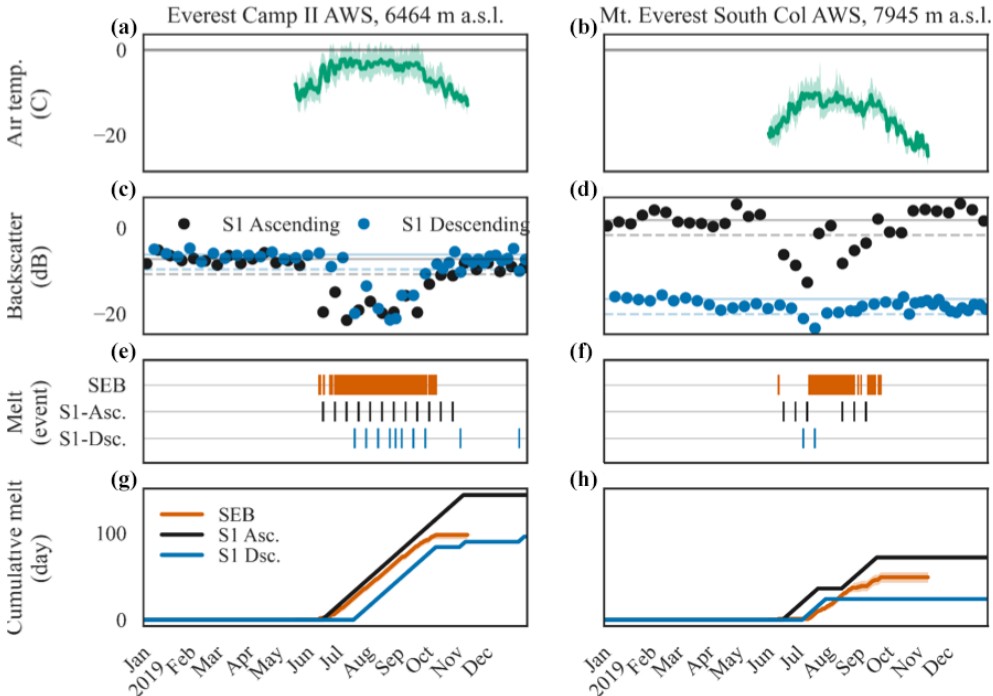

**Figure 5.** Air temperature measurements from **(a)** the Everest Camp II automated weather station (AWS) and **(b)** the Mt. Everest South Col AWS are compared to glacier surface melting observations from the Sentinel-1 satellite synthetic aperture radar (SAR). **(c)** The radar backscatter from the Khumbu Glacier (at 6483 m a.s.l.) adjacent to the Camp II AWS show a pronounced decrease in backscatter over several months associated with ongoing surface melting during summer months. Melting is identified when backscatter decreases below a threshold (dashed line), set at 3 dB below the winter mean (solid line). **(d)** At the upper reaches of the Khumbu Glacier (7128 m a.s.l.), S1-SAR observes melting during ascending passes (18:00 local time) but not during descending passes (06:00 local time), except for a brief period during late June. **(f)** Timing of surface melt from observation and SEB modeling are compared to S1 ascending and descending observations at **(e)** Camp II and **(f)** South Col AWSs. The cumulative number of melting days from the SEB model and S1-SAR are shown for **(g)** Camp II and **(h)** South Col.

lapse rates within the central Himalayas during 2018 using data from two meteorological stations within the Langtang valley (Table 1). Temperature–elevation lapse rates were determined using 3-day averages of hourly air temperature measurements interpolated to fill gaps using methods identical for the calculation of temperature–elevation lapse rates in numerical model studies of snowmelt and glacier wasting in the HKH (Baral et al., 2014). We calculated the difference between 3-day average air temperatures and divided by the difference in elevation (1148 m) between the two stations in the Langtang river valley, Nepal. Lapse rates ranged from $5\,°C\,km^{-1}$ in July of 2018 to $-13.7\,°C\,km^{-1}$ in December of the same year. Temperature–elevation lapse rates were used to extrapolate the maximum elevation of three isotherms ($-10$, $-5$ and $0\,°C$) for each day of the year in 2018 in order to compare extrapolated temperatures with melt retrievals from Sentinel-1.

## 4 Results and discussion

A melting signal ($z > 2$) is observed across all regions of significant mapped glacier area contained in the GAMDAM inventory. Melt retrievals are aggregated across 12 glacio-climate sub-regions within the HKH delineated within the HiMAP dataset (Shean et al., 2020) and averaged across the calendar years 2017–2019 to report summary statistics (Table 3). Aggregate statistics of melt onset (MO) and freeze onset (FO) are calculated across 100 m elevation bins using the 30 m SRTM (Farr, 2007) digital elevation model for each glacio-climate sub-region as presented in Fig. 4. For all sub-regions, there is a roughly linear relationship of mean MO with elevation over most ranges in elevation. The progression of MO with increasing elevation is consistent with lapse rate temperature controls on surface melting for most elevation ranges. Notably, we find an inflection toward earlier melt onset occurring at higher elevations ($>6500$ m a.s.l.). A divergence from lapse-rate-driven melting at high elevations suggests that snowmelt onset may have regional triggers, like strong solar insolation (Matthews et al., 2019) or variable

regional weather patterns, such as increases in atmospheric moisture, cloudiness and deep convection (Lau et al., 2010).

In the 3 years of freeze onset (FO) across sub-regions we do not find the level of elevation dependence as observed in MO (Fig. 6). For much of the HKH, FO occurs during a short period of time and over large spans of elevation. For example, in the central Himalaya sub-region, FO has a range of 33 d, while the MO for this region spans 79 d on average. FO across sub-regions does not follow a linear trend with elevation similar to MO (Fig. 6). In western sub-regions (eastern Hindu Kush, western Pamir and Karakoram), there is a signal of delayed refreeze apparent in summary statistics at higher elevation ranges within each respective catchment. In the western Pamir, FO at 6000 m occurs 22 d later than FO at 5000 m a.s.l (Supplement Fig. S2). Similarly, in the Karakoram, FO occurs 10 d later at 7500 m a.s.l. compared to 6500 m a.s.l. In the Tanggula Shan, FO at 6500 m a.s.l. is delayed by 21 d relative to FO at 5500 m a.s.l.

Signals of delayed refreeze are observed at elevation ranges similar to the greatest $z$ score across summary statistics of FO (Supplement Fig. S2). Notably, we find specific high-elevation ranges in select catchments in the western sub-regions (eastern Hindu Kush, western Pamir, and Karakoram) and some eastern sub-regions (Tanggula Shan, Nyainqentanglha, eastern Tibetan Mountains and Hengduan Shan), where there is a signal of delayed refreeze apparent in summary statistics. Although sub-regional aggregate FO statistics do not show delayed refreeze in larger sub-regions (i.e., the central Himalaya), we observe signals of delayed refreeze on individual glaciers within the central Himalaya, indicative of meltwater retention within percolation facies (Fig. 7). Complete refreeze across the depth of a percolation zone is delayed relative to percolation zone surfaces because liquid water is retained within a percolation zone medium CE10 after the surface of the percolation zone has frozen (Paterson, 2016). Completely frozen percolation zones produce some of the largest radar backscatter responses on the terrestrial Earth (Jezek et al., 1994). Because frozen snow and percolation facies are essentially transparent, $C$-band SAR will be sensitive to the presence of liquid water across the volume of a snowpack or firn strata (Fischer et al., 2019). Signals of delayed refreeze across sub-regions are indicative of meltwater storage within the percolation volume due to meltwater retention. A figure illustrating melt timings and $z$-score metric is included in the Supplement (Fig. S2).

## 4.1 Percolation meltwater hydrology

Delayed freeze-up apparent in summary statistics at unique elevation ranges across glacio-climate sub-regions is an important illustration of how melt retrievals from Sentinel-1 are sensitive to the presence of liquid water within the snowpack and/or firn subsurface (Brangers et al., TS9; Fischer et al., 2019). At the Khumbu Glacier on Mount Ever-

**Table 3.** TS8 Melt retrieval statistics summarized across HiMAP sub-regions and aggregated over 1 km elevation bins from the SRTM 30 m DEM (Farr, 2007). Data for each elevation bin and sub-region are structured, where the first row is the melt onset (MO) in day of year (DOY) and associated MO variance in days, the second row is freeze onset (DOY) and associated variance (days), and the third row is CE11 the area of melt retrieved in units of square kilometers.

| Elevation Range (m a.s.l.) | 3000–3999 | | 4000–4999 | | 5000–5999 | | 6000–6999 | | 7000–7999 | |
|---|---|---|---|---|---|---|---|---|---|---|
| Central Himalaya | 90 (21.3) | 49 | 274 (14.9) | 1492 | 134 (20.1) | 7374 | 268 (15.8) | 2303 | 169 (13.7) | 153 |
| | 277 (22.7) | | 104 (17.1) | | 274 (17.6) | | 156 (12.9) | | 249 (14.7) | |
| Eastern Himalaya | 75 (17.) | 25 | 292 (24.9) | 353 | 129 (18.9) | 2621 | 279 (18.6) | 774 | 161 (12.6) | 22 |
| | 293 (46.6) | | 102 (20.6) | | 290 (20.3) | | 152 (13.2) | | 260 (12.3) | |
| Eastern Hindu Kush | 101 (13.1) | 89 | 257 (27.1) | 2179 | 152 (12.) | 1333 | 243 (10.8) | 177 | 175 (20) | 11 |
| | 276 (42.8) | | 131 (17.2) | | 253 (16.3) | | 180 (13.6) | | 243 (13.3) | |
| Eastern Tibetan Mountains | – | – | 268 (12.8) | 26 | 275 (15.3) | 381 | 280 (5.7) | 4 | – | – |
| | – | | 136 (10.5) | | 151 (10.4) | | 173 (7.3) | | 258 | |
| Gangdise Mountain | – | – | – | – | 1123 | | 438 | 177 | – | – |
| | – | | – | | 247 (11.3) | | 250 (9.1) | | – | |
| Hengduan Shan | 90 (27.8) | 11 | 294 (21.9) | 231 | 289 (24.6) | 1245 | 289 (16.9) | 43 | 151 (5.3) | 1 |
| | 294 (39.4) | | 113 (17.7) | | 121 (24.5) | | 146 (15) | | 269 (7) | |
| Karakoram | 105 (19.4) | 955 | 268 (19.1) | 5889 | 262 (19.4) | 15060 | 249 (13.8) | 2752 | 171 (20.5) | 110 |
| | 278 (23.8) | | 128 (20.1) | | 152 (15.1) | | 170 (17.3) | | 248 (12.3) | |
| Nyainqentanglha | 79 (19.9) | 145 | 310 (18.1) | 2637 | 302 (24.7) | 5253 | 294 (22.2) | 295 | 167 (9.1) | 1 |
| | 300 (38.6) | | 105 (13.9) | | 131 (18.3) | | 150 (11.3) | | 269 (4.3) | |
| Tanggula Shan | – | – | 290 (14.2) | 17 | 265 (20.4) | 2184 | 284 (29.2) | 112 | – | – |
| | – | | 120 (13.7) | | 161 (13.5) | | 169 (9.) | | – | |
| Tibetan Interior Mountains CE12 | – | – | 299 (11.7) | 3109 | 249 (14.1) | 172 (12.4) | 249 (12.) | 1634 | – | – |
| | – | | 136 (.6) | | 166 (11.3) | | – | | – | |
| Western Himalaya | 94 (9.3) | 156 | 275 (15.6) | 3651 | 263 (22.1) | 6135 | 247 (13.3) | 357 | 162 (18.1) | 13 |
| | 280 (22.8) | | 113 (14.8) | | 138 (20.5) | | 163 (11.8) | | 250 (10.9) | |
| Western Pamir | 117 (20.7) | 602 | 255 (23.1) | 5271 | 250 (16.4) | 4103 | 243 (12.) | 117 | 170 (9.2) | 2 |
| | 275 (28.5) | | 141 (16.5) | | 159 (12.7) | | 177 (13.2) | | 249 (8.7) | |

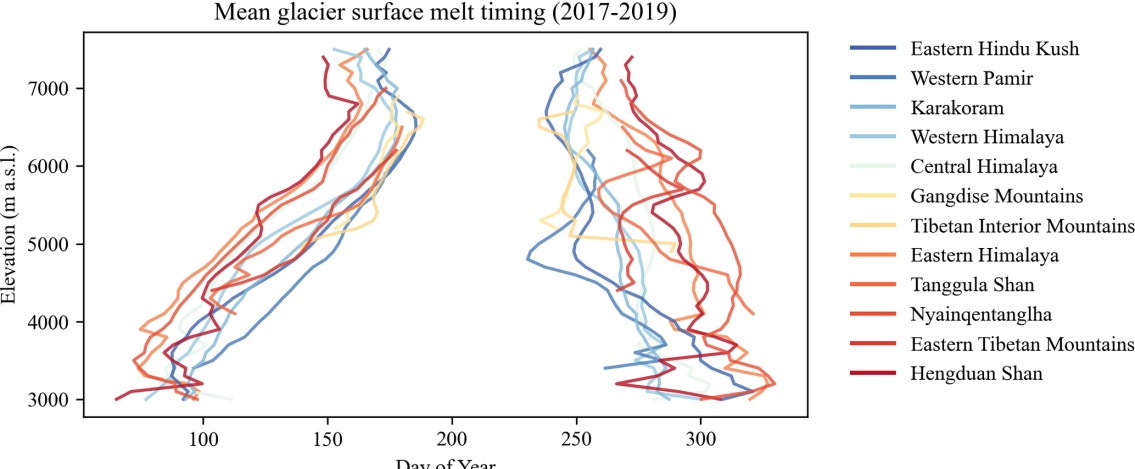

**Figure 6.** Mean melt onset (MO; left) and freeze onset (FO; right) summarized in 100 m elevation bins using the 30 m SRTM digital elevation model (Farr, 2007) and 12 HiMAP sub-regions (Shean, 2020). The blue-to-red color scale indicates the longitude of the HiMAP region centroid, where the westernmost regions are shown in dark blue and easternmost shown in dark red.

est, Sentinel-1-retrieved refreeze occurs over 30 d later at $\sim 6000$ m a.s.l. compared to elevations below 5400 m a.s.l. and above 6200 m a.s.l., indicating that liquid meltwater was retained at elevation ranges between $\sim 5400$–6200 m a.s.l. during a month when elevations both above and below this range were recorded as completely frozen within Sentinel-1-retrieved melt signals. The time series of mean Sentinel-1 SAR backscatter for descending orbital nodes from two 250 m buffered points on the Khumbu Glacier show a rapid increase in SAR backscatter magnitude for the higher-elevation location, whereas backscatter time series extracted from within the elevation range of delayed melt offset show a gradual increase in radar backscatter. We interpret this gradual backscatter increase to be indicative of gradually decreasing liquid water content in the snowpack (or firn) as refreeze progresses from the glacier surface and into the depth of the percolation zone (Fig. 7) (TS10 Forster et al., 2014; TS11 Miège et al., 2016). This elevation range ($\sim 5400$–6200 m a.s.l.) is similar to known elevation ranges of percolation zones on the Khumbu Glacier as detailed in recent fieldwork (Matthews et al., 2019, 2020). SAR backscatter time series showing a gradual increase in backscatter within regions of known percolation suggest that there is a relationship between frozen percolation zone depth and the rate of $C$-band backscatter change across refreeze cycles. It has been shown that $C$-band backscatter gradually increases with frozen percolation zone depth and decreasing percolation zone wetness during a re-freeze process (Ashcraft and Long, 2005).

## 4.2 Spatial variability: radar scattering and glacier facies

Imaging radar backscatter intensity and response to surface melting are linked with glacier facies (Ramage et al., 2000;

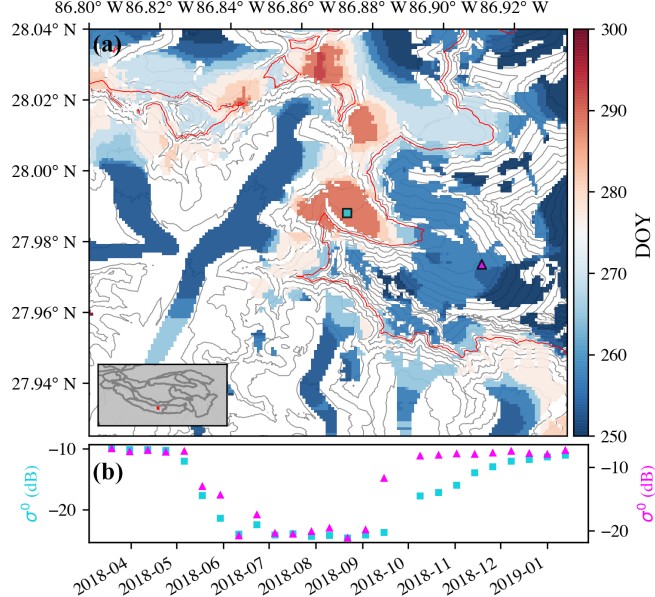

**Figure 7.** (a) Refreeze timing over the Khumbu Glacier region of Mount Everest in the central Himalaya. Red regions of freeze onset occur at mid-elevations, indicative of delayed refreeze due to melt-water retention in percolation zones. (b) Sentinel-1 backscatter time series from two points on the Khumbu Glacier, one within known elevations of glacier percolation facies (teal square; 6000 m a.s.l.) and another point at elevations where temperatures likely do not exceed 0 °C annually (pink triangle; 6600 m a.s.l.).

Rau et al., 2000; Zhou and Zheng, 2017). Snow melting on the glacier surface produces a strong decrease in radar backscatter across all glacier facies. In the accumulation zone the refreeze signal is also pronounced as the dissipation of strongly absorbing wet snow at the surface is followed by

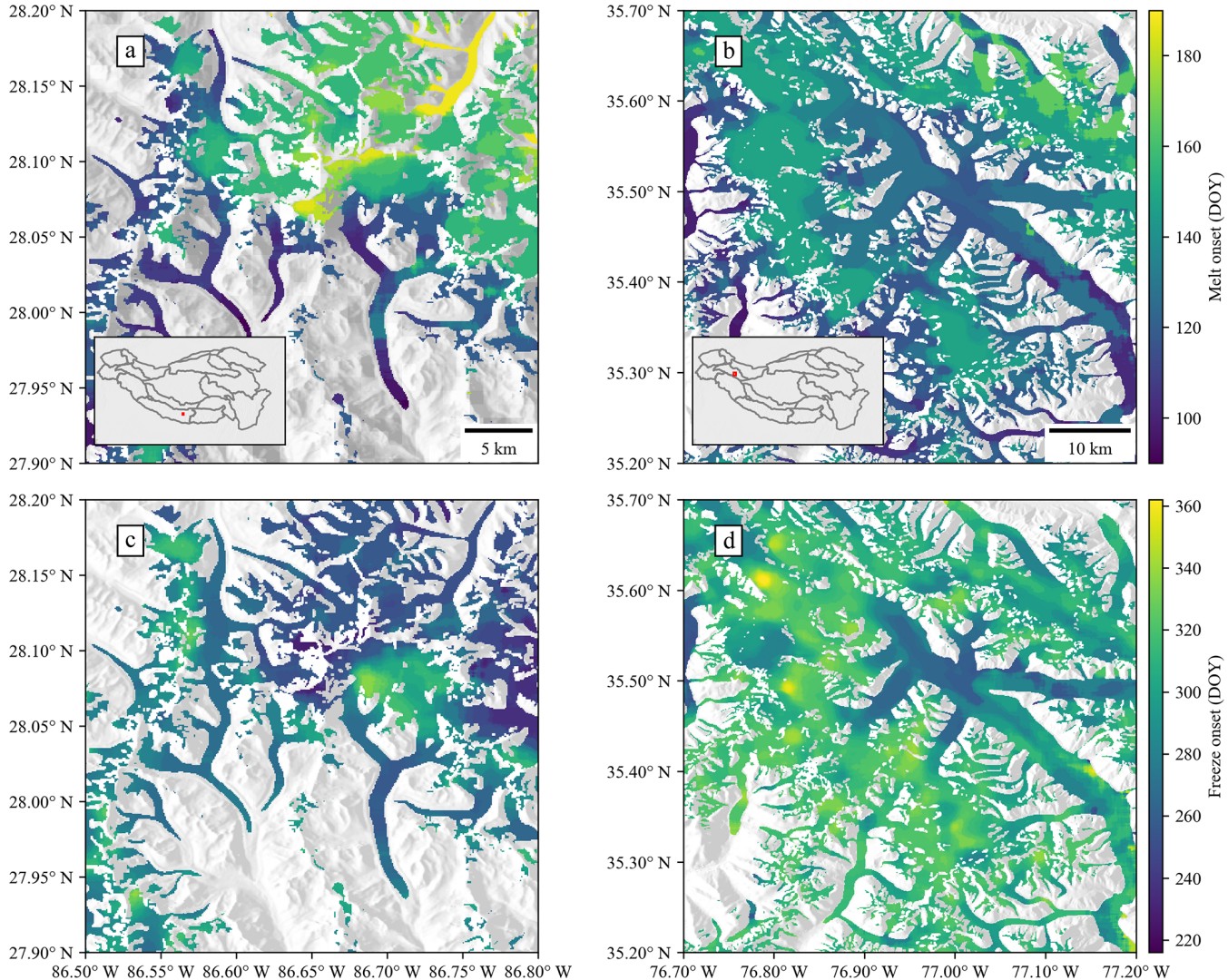

**Figure 8.** Melt retrievals averaged over the calendar years 2017–2019 in the central Himalaya and Karakoram regions. **(a)** Mean melt onset (DOY) in the central Himalaya. **(b)** Mean melt onset (DOY) over the Siachen Glacier in the Karakoram region. **(c)** Mean freeze onset (DOY) in the central Himalaya. **(d)** Mean freeze onset (DOY) over the Siachen Glacier in the Karakoram region. Data overlay a 30 m Shuttle Radar Topography Mission (SRTM) DEM hillshade (Farr, 2007).

volume scattering from deep snowpack and stratified ice layering. The scattering response to refreeze in the ablation zone is more complex and not well characterized. Here, supraglacier features like crevasses, suncups, debris cover and other heterogeneities are likely to cause highly variable radar scattering mechanisms over short distances upon the disappearance of seasonal snow from the ablation surface (Rott and Mätzler, 1987). We use the $z$-score metric to select areas where radar backscatter increases substantially during the refreeze process. However, since scattering response during the transition from wet snow will differ with various surface features (e.g., bare ice, debris and supraglacier ponding), it is difficult to isolate the refreeze response. Average $z$ is minimum in the HKH across the lowest-elevation glacier sur-

faces (2000–4000 m a.s.l.), whereas $z$ is maximum at unique elevation ranges within sub-regions (Figs. 4, S2). Ablation zone surfaces (at lower elevations) do not exhibit the magnitude of backscatter intensity of percolation zones, and therefore lower glaciated elevations show lesser seasonal contrast in backscatter compared to higher elevations. These differences are also apparent in the spatial granularity of melt retrievals from the S1-SAR product, as shown in Fig. 8. Ablation zone surfaces on valley glaciers show spatial heterogeneity in MO indicative of supraglacial features, like debris cover rather than randomly distributed noise. There exists uncertainty in the FO signal on glacier ablation surfaces that will require further investigation. In ablation areas with lower sensitivity to melting, we hypothesize that snow-off condi-

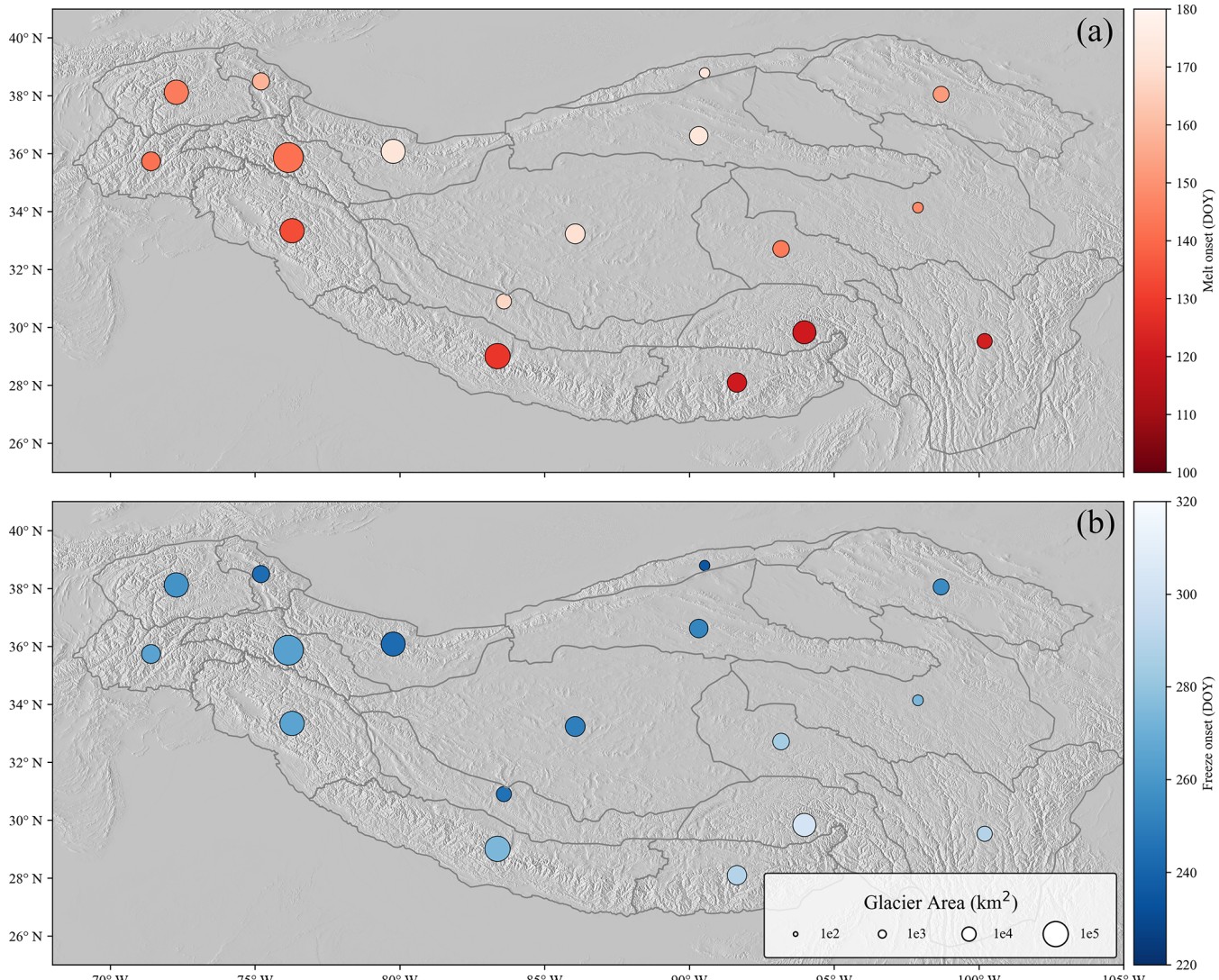

**Figure 9.** Melt onset **(a)** and freeze onset **(b)** averaged over 2017–2019 plotted over an SRTM 30 m DEM hillshade (Farr, 2007). Melt retrievals are averaged across HiMAP glacio-climate sub-regions (Bolch et al., 2019a; Shean et al., 2020) and scaled by the mapped glacier area within each sub-region.

tions result in brightening of the radar signal due to surface scattering contributions from wet debris, bare ice or other ablation surface heterogeneities. For this reason, at lower elevations where annual air temperatures exceed 0 °C (i.e., where temperature–elevation lapse rates hold), lapse rate estimates of elevation might be more robust estimates of FO using this approach. Overall, surface melting signals appear to be consistent with expectations of temperature lapse rates (i.e., earlier melting and later refreeze at lower elevations) across elevations where annual air temperatures likely exceed 0 °C (<6000 m a.s.l.). We have illustrated the spatial granularity of melt retrievals in Fig. 8 in addition to average melt onset and offset by sub-region in Fig. 9.

## 4.3  Considerations of temperature–elevation lapse rates

We compare SAR retrievals of MO and FO to temperature–elevation lapse rates derived within a catchment in the central Himalaya to investigate SAR retrievals alongside lapse rate assumptions of glacier melt status using methods and AWS data for the construction of lapse rates from prior studies in the Langtang valley, central Himalaya (Baral et al., 2014). In 2018, we observe that the average MO is found to follow the 0 and −5 °C isotherms for elevations ∼ 4500 to 6500 m a.s.l. (Fig. 10). Below and above these elevations and for FO, we find episodic melting events occurring over a range of elevations. This is especially apparent in the FO around day of year 270, where FO occurs within a roughly 2-week period

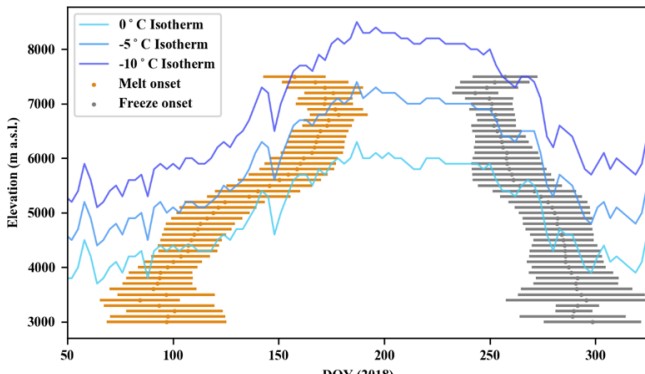

**Figure 10.** Sentinel-1 SAR-retrieved melt onset (orange) and freeze onset (gray), with spatial variability at $\pm 1$ standard deviation, across the central Himalaya region. The elevations of the 0, $-5$ and $-10\,°C$ isotherms from 2018 are overlaid for comparison. Melt signals are recorded in excess of 3 months at elevations extending $>1$ km above the maximum elevation of the $0\,°C$ isotherm, indicative of a sustained presence of liquid water within the snow matrix across these high-elevation ranges.

across glaciers between 5000–7500 m a.s.l. MO and FO signals are retrieved on days and at elevations where lapse-rate-derived temperatures do not exceed $-10\,°C$, which strengthens and expands recent in situ observations on glacier melt at the Khumbu Glacier in the Mount Everest region showing that incident shortwave radiation drives melt at these temperatures and elevations (Matthews et al., 2019). Here we observe that, even at these extreme elevations ($>7000$ m a.s.l.), melt signals persist for over 3 months on average across the central Himalaya, which suggests that liquid water is retained at these high elevations across a seasonal melt cycle and may not be hydrologically negligible. In radar-derived observations there is a discrepancy between SAR and lapse-rate-estimated melting records that occurs at elevations extending 1 km above the maximum $0\,°C$ isotherms in the central Himalaya. Glaciated areas in the central Himalaya at elevations greater than 6000 m a.s.l. – the approximate maximum elevation of the $0\,°C$ isotherm for 2018 – account for 21.58 % (2,453 km$^2$) of total glaciated area within the region.

### 4.4 Melt retrievals and glacio-climate sub-regions

The 3-year record of Sentinel-1 SAR retrievals of glacier melt status represents a baseline measurement for the HKH region. The summary melt statistics are aggregated over HiMAP sub-regions in order to compare melt retrievals and sub-regional estimates of glacier mass loss (Shean et al., 2020). Overall, the HKH sub-regions with the most rapid mass loss between 2000–2010 tabulated in Shean et al. (2020) (eastern Himalaya, Hengduan Shan, Nyainqên-tanglha) exhibit the greatest number of melt days on average in 2017–2019 from Sentinel-1 retrievals. Sub-regions with slower mass loss (eastern Hindu Kush, western Pamir,

Karakoram, Tibetan interior) show on average 1 month less of melt duration relative to regions with accelerated mass loss. Interestingly, the Gangdise sub-region, with one of the higher post-2000 rates of glacier wasting in the HKH, shows annual melt durations of less than 3 months on average, which appears more characteristic of western regions with slower mass loss (i.e., Tibetan Interior Mountains). Although Sentinel-1 retrievals of glacier melt status for 3 calendar years do not make up a climatic record, we observe that between 2017–2019 there was on average less duration of melting in regions where in situ data and climate models indicate that frozen winter precipitation contributes to glacier accumulation despite warming global climate (Karakoram, Hindu Kush, eastern Pamir, western Himalaya) (Kääb et al., 2015; Kapnick et al., 2014; Palazzi et al., 2013). We interpret shorter duration of annual melt days in the western regions of the HKH as a potential indicator of the "Karakoram anomaly" reflected in the Sentinel-1 data record. Because the meteo-climatic drivers of the Karakoram anomaly are still under debate (Farinotti et al., 2020), Sentinel-1 retrievals of melt duration may be useful for interrogating meteo-climatic drivers of heterogeneity in glacier wasting dynamics across the HKH.

### 5 Conclusions

Synthetic aperture radar time series backscatter images and glacier extent maps derived from optical imagery have long been proposed to inform hydrologic and glaciologic research across the global cryosphere; however a harmonized dataset of glacier surface melt does not exist. We retrieve glacier surface melt timing and duration for the study years 2017–2019 across the HKH region using time series $C$-band SAR from the Sentinel-1 A/B satellites and an inventory of 105 432 glaciers spanning 83 102 km$^2$ of ice-covered area. We quantify the magnitude of the seasonal melt signal by comparing mean summer and winter backscatter using a $z$-score metric and retrieve constraints on seasonal melt characteristics across all glaciated elevations of HKH at 90 m spatial and 12 d temporal resolution. Melt conditions in surface energy balance models of glacier melt, driven by in situ meteorological data from Mount Everest, fall within the date ranges of melt retrievals recorded in Sentinel-1 SAR data. Comparison of melt retrievals to temperature–elevation lapse rates calculated using two high-elevation meteorological stations in the central Himalaya reveals that melt onset persists for over 3 months at elevations where extrapolated air temperature fields do not exceed $-10\,°C$. Melt is retrieved across all elevation ranges of HKH glaciers, which suggests that a dry-snow accumulation zone in the HKH region does not exist. Meltwater retention is indicated within known glacier percolation zones on Mount Everest through signals of delayed refreeze. Delayed refreeze occurs across the HKH at elevations with the greatest seasonal contrast in backscatter

intensity, attributable to radar scattering in percolation facies. Melt signals persist for a greater portion of the year in regions known for rapid contemporary glacier wasting (i.e., central and eastern Himalaya sub-regions), whereas regions with a more stable glacier mass balance (i.e., Karakoram) exhibit a shorter duration of annual melt. We produce a geospatial data product of melt onset (DOY) and freeze onset (DOY) spanning glaciers of the HKH region at 90 m spatial resolution for the calendar years 2017–2019 and plan to release annual updates to this dataset each calendar year across the mission duration of Sentinel-1. The methods presented in this study can provide the basis for an operational monitoring system of glacier surface melt dynamics and aid the development and assessment of surface energy balance models of glacier ablation across the global cryosphere.

*Code and data availability.* The data are available from the National Snow and Ice Data Center here: https://doi.org/10.5067/05I6ZHZWHSVV TS12 (Steiner et al., 2021). The code used to produce the data is available here: https://github.com/porefluid/glacier_melt TS13 .

*Supplement.* The supplement related to this article is available online at: https://doi.org/10.5194/tc-15-1-2021-supplement.

*Author contributions.* NCS and KCM devised the project and the main conceptual ideas. CS developed and executed the 425 final methodological approach CE13 and authored the computer code. CS contributed most of the writing to the manuscript, with major contributions from NCS. KCM supervised the project and manuscript.

*Competing interests.* The authors declare that they have no conflict of interest.

*Acknowledgements.* This work was supported by funds provided to The City College of New York by the National Aeronautics and Space Administration Cryosphere program's High Mountain Asia Team (HiMAT) program, under award number NNX16AQ83G. Portions of this work were conducted at the Jet Propulsion Laboratory, California Institute of Technology, under contract to the National Aeronautics and Space Administration.

*Financial support.* TS14 This research has been supported by the National Aeronautics and Space Administration Cryosphere program (grant no. NNX16AQ83G).

*Review statement.* This paper was edited by Melody Sandells and reviewed by three anonymous referees.

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

## Remarks from the language copy-editor

CE1    Does the slash here represent "and" or "or"? Please confirm for all instances.

CE2    Is melt/freeze state the detection classification? Please clarify this relationship.

CE3    Please confirm.

CE4    Please define.

CE5    It is our house standard to pluralize abbreviations with an "s". Please confirm that the singular and plural forms are used correctly throughout the paper.

CE6    Does the slash here represent "and" or "or"?

CE7    What is being illustrated?

CE8    This range is not in our reference atlas. Are you referring to mountains in (eastern) Tibet in general? Please check.

CE9    Please confirm.

CE10    Please confirm.

CE11    Please confirm.

CE12    This range is not in our reference atlas. Are you referring to interior mountains in Tibet in general? Please check.

CE13    Does 425 refer to the number of approaches?

## Remarks from the typesetter

TS1    The composition of Figs. 1, 4–5, and 7–9 has been adjusted to our standards.

TS2    Please provide year.

TS3    Please note that after a section there should always be more than one subsection and after a subsection more than one subsubsection. If not, there should be no numbering for the one subsection/subsubsection.

TS4    Do you mean "2018"?

TS5    Please check and confirm new section numbering throughout the paper. The section "Methods" had no numbering in the submitted PDF. Please adjust/change also the according sections in the text if necessary.

TS6    Please provide reference entry.

TS7    Please confirm new sectioning.

TS8    Please check and confirm new formatting of the table. There was something wrong with the Table Tool in the submitted WORD/PDF document and therefore the transformation from DOCX into TEX caused some problems.

TS9    Please provide year.

TS10    Please provide reference entry.

TS11    Please provide reference entry.

TS12    Please check DOI. Link is not working.

TS13    Please provide a reference list entry including creators, title, and date of last access.

TS14    Please note that there is a discrepancy between funding information provided by you in the acknowledgements and the funding information you indicated during manuscript registration, which we used to create this section. Please double-check your acknowledgements to see whether repeated information can be removed from the acknowledgements or changed accordingly. If further funders should be added to this section, please provide the funder names and the grant numbers. Thanks.

TS15    Please ensure that any data sets and software codes used in this work are properly cited in the text and included in this reference list. Thereby, please keep our reference style in mind, including creators, titles, publisher/repository, persistent identifier, and publication year. Regarding the publisher/repository, please add "[data set]" or "[code]" to the entry (e.g., Zenodo [code]).

TS16    Please provide pages.

TS17    Please provide journal name.

TS18    Please provide place of publication.

TS19    Please provide jorunal name, volume and pages.

TS20    Please provide volumem, pages and year.

TS21    Please provide pages.

TS22    Please provide URL link and more information like creator, title, what kind of data, etc. It is not usual to cite only a searching provider. Or can this citaiton be deleted here and in the text?

TS23    Please check initials.

TS24    Please provide journal name.

TS25    Please provide pages.

TS26   Please provide pages.
TS27   Please provide pages.
TS28   Please provide pages.
TS29   Please check initials.
TS30   Please provide journal name.
TS31   Please provide DOI.
TS32   Please provide DOI.
TS33   Please provide pages.
TS34   Please provide pages.
TS35   Please provide DOI.
TS36   Please provide journal name, volume, and pages.
TS37   Please provide volume and pages.
TS38   Please provide place of publication.
TS39   Please provide pages.
TS40   Please provide publisher.
TS41   Please provide journal name and volume.
TS42   Please provide palce of publication.
TS43   Please check editor.
TS44   Please provide pages.
TS45   Please check DOI link and see comment above in the data availabilty section.
TS46   Please provide pages.
TS47   Please provide pages.
TS48   Please check and confirm initials.
TS49   Please provide journal name.
TS50   Please provide pages.
TS51   Please provide pages.
TS52   Please provide pages.