# Peer review of "Mapping seasonal glacier melt across the Hindu Kush Himalaya with time series SAR"

_The Cryosphere, 2020_

## Referee Comment (RC1) · Anonymous Referee #1 · 4 Oct 2020

**General comments**

This manuscript discusses the use of time series of Sentinel-1 SAR imagery to map regional melt characteristics of glacier ablation in the Hindu Kush Himalaya region. The topic is of high importance to further understand how to operationally use Sentinel-1 SAR backscatter intensity for mapping glacier characteristics. The authors are using the SAR data to investigate the duration of seasonal glacier melt, these are important inputs to surface energy models. Additionally, using time series SAR data is of high importance for further understanding of how to map glaciers using SAR backscatter signals, e.g. refreezing of liquid water in the percolation zone.

The authors give a good overview of the regional glacier melt dynamics in the HKH region and this is a very interesting study. However, the presented results are rather general in terms of the regions studied and lacks a proper error analysis. The text has some long and complex sentences here and there, but is overall relatively easy to understand.

**Specific comments**

The chapter about SAR processing (2.3 Computer Infrastructure) is too limited, e.g. there are many processing options in SNAP that are not described properly. To be able to reproduce the data, more information about the processing of Sentinel-1 satellite images is thus needed. The authors should also be aware of the limitations of the SRTM DEM over glaciers used here for processing the Sentinel-1 data (e.g. Kääb, A., Treichler, D., Nuth, C., and Berthier, E.: Brief Communication: Contending estimates of 2003–2008 glacier mass balance over the Pamir–Karakoram–Himalaya, The Cryosphere, 9, 557–564, https://doi.org/10.5194/tc-9-557-2015, 2015.).

Unfortunately, the authors do not give detailed error estimates in relation to the Sentinel-1 SAR time series over glacier ablation. The study lacks in situ observations for comparison with the SAR backscatter intensity, and the authors should discuss whether the results are trustworthy in the HKH-region. Even though in situ measurements might be absent, the authors might compare the melt signals retrieved from Sentinel-1 SAR data to other remote sensing sources (optical e.g. Landsat-8 and/or Sentinel-2 and SAR e.g. Radarsat-2).

The four glacier regions studied are rather large, and it would have been interesting to understand more of the local variations of backscatter intensity values within these regions. A suggestion is to pick out some glaciers from each region and compare results from these individual glaciers.

If possible, the authors should discuss the refreezing process in the percolation zone in the Hindu Kush Himalaya region in more detail. Are there more examples of the results described in Figure 8?

L 70, P 3: "This study builds on extensive research into microwave scattering from dry and wet snow and techniques for snowmelt retrieval from imaging radar sensors to present an operational monitor for spatially-resolved glacier surface melt characteristics using synthetic aperture radar (SAR) time series and up-to-date glacier outlines derived from satellite optical imagery across the HKH." Which years are the up-to-date glacier outlines from? (Sakai, 2019)

L 85, P 3: "In areas that are seasonally snow-free, like for areas of debris-cover or bare ice, melting conditions are dominated by surface scattering that is significantly darker relative to winter conditions (Lievens et al., 2019)." What is meant by "darker"? Is this connected to roughness differences between debris-cover, bare ice and wet snow showing different backscatter intensity?

L 95, P 4: "Limited only by the frequency of observations (12-days per orbit direction)". Are 6-days repeat orbits of Sentinel-1 available in the study region? If not, will 6-days repeat be available in the future? If so, this should be discussed.

L 155-160, P 7: Why only Sentinel-1 SAR imagery from 2017-2019? Sentinel-1A was operative since 2014.

L 160, P 7: "By combining orbit directions, we utilize observations acquired at day and night. For the purpose of this study we do not attempt to resolve diurnal-scale melt freeze processes and instead focus on retrieving annual characteristics of melt timing and duration.". The backscatter intensity signal differs between day and night. Further explanation on how this might affect the results must be included.

L 261, P11: "Mean seasonal melt magnitude averaged over 100m elevation bins over all three calendar years of data shows strong (z > 2) melt signals across glacio-climatic sub-regions and across all elevation ranges of significant glaciation (Fig. 4). The occurrence of seasonal melt signals across all ranges of elevation in the HKH is both noteworthy and striking." These statements should be placed in the Results and Discussion section.

L 315, P 14: "A summary of glacier melt timing with elevation averaged across study years is shown across 100m elevation bins in Figure 5 and tabulated across 1km elevation bins in Table 2. Due to errors in melt classification at elevations below 3,500m a.s.l., we will summarize our observations across elevations >3,500m a.s.l. and assume that the relatively small glaciated area at elevations at or below 3,500m a.s.l. is negligible for the purposes of identifying trends across regions and elevation." What kind of errors are referred to? And what are the assumptions based on?

**Technical comments**

Figure 1: Top: Glacier outlines are not that apparent in the figure. Bottom: Blue color of descending swaths would be clearer.

Figure 2: Figures are too small and dark, and are hard to interpret. A suggestion is to cut down on the amount of figures and enlarge the most important ones that indicates VH polarization which is used in this work. Consider a brighter color on glacier outlines.

Figure 3: Use same scale in the plots for VV and VH backscatter.

Figure 5: Suggestion to have elevation on the y-axis. Make the plots broader so more information can be interpreted. Include stippled lines for each 50 DOY in the figure to help the reader to interpret the results.

Figure 6: Difficult to understand and interpret the small circular insertions in the plot. Make them larger and explain more carefully, if these are of big importance.

Figure 8: Suggest to show the location on a map and include glacier outlines.

References should be in chronological order. e.g. L72, P 3.

---

## Referee Comment (RC2) · Anonymous Referee #2 · 5 Oct 2020

This paper provides an application of C-Band Sentinel-1 imagery to map seasonal melting along the Hindu Kush Himalaya (HKH) for the period 2017-2019. The paper provides spatiotemporal time series of several metrics, as for example freeze onset, for the region. The authors detected horizontal (at the regional scale) and vertical differences that they assess resemble well established HKH glacio-climatic regimes while providing insights on scarcely described dynamics, such as the occurrence of melt signals at very high elevations.

My general assessment is that the paper is straight: it describes a method that naturally connects with the results. The application by itself is of great value in other regions for monitoring as well as for model validation, given the high resolution of the imagery and the mission expected length of 7 years. After reading the paper several times, however,

[Figure]

I am not fully convinced the paper fits in this journal in its present form instead of a more remote sensing oriented journal. I think it lacks of a more in depth glacier-climate interpretation. Below I include some comments that argument my assessment.

General comments

(1) The paper provides a detailed description of the methods but to me the section "Results and Discussion" lacks discussion on glacier-climate regimes. The authors claim that melt dynamics coincide with different HKH glacio-climatic regimes. As far as I can tell, these regions correspond to operational inventory subdivisions (for CGI or RGI) and do not necessarily correspond to boundaries between well-defined glacier regimes. Perhaps just rearranging the narrative will clarify this, but I would suggest to expand the discussion including some reasons why these areas show these differences or whether other regional differentiation is possible.

(2) Along the same lines, I believe the paper needs to add some sort of longer climatic discussion. Currently, lines 418-419 present a reworked version of 131-132. Is it possible for the authors to include some literature or analysis on the possible weather patterns that would explain the findings for the studied period? Perhaps looking at re-analysis fields for the MO and/or FO days will allow the authors to further elaborate on the findings. In addition, this can be helpful in trying to explain the high elevation melting events.

(3) In line 374 I'm not sure if longwave energy is what drives melt at these elevations. I think that the Everest climate data is showing that the large input of shortwave energy and monsoonal activity allow for the glacier surface to reach melting temperatures despite below freezing air temperatures. Since the authors are seen this possible effect, I think there is a great opportunity to test whether some of the high-elevation melt events coincide with the conditions depicted in Matthews et al (2020), by looking at weather conditions around those dates.

(4) L184: Can you provide more details on the methods included in the computing

infrastructure?

(5) L206: Is it possible to attempt a sensitivity analysis of the b value? Could the choice of the b value be partially contributing to uncertainty in the results?

(6) Figure 3: I think the validation data is insufficient to make general statements on the method for the whole region. I wonder if it is possible to make a comparison relative to reanalysis air temperature at equivalent geopotential levels for a larger region.

(7) Figure 5: wouldn't it be more straightforward to leave elevations in the y axis and DOY in the x axis? Also, although the spread of DOY do not seem to be statistically different there is an interesting divergence between about 5700 to 6700, where Karakoram and Western Himalaya tend to cluster differently relative to Central and Eastern Himalaya. Perhaps studying some air temperature and humidity dynamics at the corresponding geopotential levels will allow the authors to interpret that situation.

References Matthews, T., and Coauthors, Going to Extremes: Installing the World's Highest Weather Stations on Mount Everest. Bull. Amer. Meteor. Soc., doi: https://doi.org/10.1175/BAMS-D-19-0198.1.

––––––––––––––––––––––––––––––

---

## Author Comment (AC1) · 30 Nov 2020

**Reviewer 1: General comments**

This manuscript discusses the use of time series of Sentinel-1 SAR imagery to map regional melt characteristics of glacier ablation in the Hindu Kush Himalaya region. The topic is of high importance to further understand how to operationally use Sentinel-1 SAR backscatter intensity for mapping glacier characteristics. The authors are using the SAR data to investigate the duration of seasonal glacier melt, these are important inputs to surface energy models. Additionally, using time series SAR data is of high importance for further understanding of how to map glaciers using SAR backscatter signals, e.g. refreezing of liquid water in the percolation zone.

The authors give a good overview of the regional glacier melt dynamics in the HKH region and this is a very interesting study. However, the presented results are rather general in terms of the regions studied and lacks a proper error analysis. The Text has some long and complex sentences here and there, but is overall relatively easy to understand.

We thank the referee for careful review of our work and thoughtful suggestions for improvement. We have conducted major revisions detailed in the responses below. We include all updated figure captions in the response text.

**Specific comments**

The chapter about SAR processing (2.3 Computer Infrastructure) is too limited, e.g. there are many processing options in SNAP that are not described properly. To be able to reproduce the data, more information about the processing of Sentinel-1 satellite images is thus needed.

**Author response (AR) 1:** We agree that the section was limited and will modify Section 2.3 as follows; and welcome any further suggestions:

**Computing Infrastructure**

A cloud-computing platform and application programming interface with pre-processed radiometrically terrain corrected Sentinel-1 A/B data was used to detect melt characteristics across the region (Gorelick, Hancher et al. 2017). Radiometric terrain
correction of Sentinel-1 data was conducted upon ingestion to the cloud server using the ESA's method contained within the Sentinel Applications Platform (SNAP) processing toolbox (ESA). The SNAP toolbox is used for processing of Sentinel-1 images and through a stepwise process of updating orbit metadata with restituted orbit files, removal of invalid edge data and low intensity noise, removal of thermal noise, computation of backscatter intensity, and finally orthorectification (Google 2020). The SNAP toolbox terrain correction functionality utilizes the 30m spatial resolution SRTM DEM (Farr 2007, Margulis, Liu et al. 2019). The pre-processed SAR times series data and API functionality used to derive glacier melting characteristics are available from GEE and can be used to recreate the work presented in this study.

The authors should also be aware of the limitations of the SRTM DEM over glaciers used here for processing the Sentinel-1 data (e.g. Kääb, A., Treichler, D., Nuth, C., and Berthier, E.:Brief Communication: Contending estimates of 2003–2008 glacier mass balance over the Pamir–Karakoram–Himalaya, The Cryosphere, 9, 557–564, https://doi.org/10.5194/tc-9-557-2015, 2015.).

**AR2:** We are aware of the bias in SRTM DEMs caused by the depth of penetration using C-band observations to determine the surface height. Additional errors associated with using SRTM in complex terrain extend beyond C-band penetrations. More error is associated with shadowing and foreshortening. Hence the gap filling performed using the SRTM 30m DEM. However, the radar signal associated with melting is large and well above any variability in normalized backscatter from errors in terrain correction. Since we are selecting time-series with constant viewing geometry, not normalizing to a constant incidence, the errors from SRTM should be consistent within each pixel. We acknowledge that by using elevation bins from the SRTM DEM to calculate summary statistics over climate subregions of the HKH our results may be biased towards higher elevations. Based on prior work, we estimate this bias will be less than

**TCD**
Reviewer Comment: Unfortunately, the authors do not give detailed error estimates in relation to the Sentinel-1 SAR time series over glacier ablation. The study lacks in situ observations for comparison with the SAR backscatter intensity, and the authors should discuss whether the results are trustworthy in the HKH-region.

AR3: We agree with the reviewer's comment, it would be ideal to provide a detailed and spatially resolved estimate of error for our surface melting observations. However, it is a well known problem that observational data at high elevations, and especially in High Mountain Asia (HMA), are severely lacking. For this reason it is vital that remote sensing is used to address deficiencies in observation, rather than being reliant on surface stations. We contend that time series synthetic aperture radar (SAR) instruments are ideally suited to measure glacier melting, in some cases better than using weather station measurements such as air temperature alone. We provide evidence of this point below.

There is an extensive record of research focused on radar applications to surface melting observation over glaciers and ice-sheets in polar regions (e.g., (Shi and Dozier 1995, Ashcraft and Long 2001, Ashcraft and Long 2007, Trusel, Frey et al. 2013, Steiner and Tedesco 2014)). It is clear that surface melting over glaciers produces a very strong and very well understood radar signal that is well established in radiative transfer theory (Ulaby and Stiles 1980). As indicated in the manuscript Figure 4 (Figure AR2, below) this melting response is measured commonly at 20x the variance of the baseline reference state (winter) and at almost always over 5x. Second, backscatter signals associated with glacier melting are unambiguous in time series observations over ice and snow, there are few seasonal phenomena that create confusion in radar melt-detection approaches. Because of this, SAR time-series observations of surface
melting are used by Johnson et al. (2020) as sources of validation and calibration for surface melting estimates in areas where surface measurements are not practical (Johnson, Fahnestock et al. 2020).

To better demonstrate the utility of radar backscatter for characterizing surface melting we have added a comparison with newly-available surface meteorology measurements from the Mt. Everest region. Automated weather stations (AWS) were installed by the National Geographic Everest Expedition and detailed in Matthews, et al. (2020) (Matthews, Perry et al. 2020). We estimate periods of surface melting from the full surface energy balance (SEB) at the surface using these stations (Figure AR1). We find that Sentinel-1 melt classifications detect SEB estimates of surface melting at the Everest Camp II AWS (6464m a.s.l.) and with an accuracy score that ranges from 73 to 85 percent and from 63 to 68 percent at Mt. Everest South Col (7945m a.s.l.) depending on the parameterization of surface roughness used in SEB estimates of melting. In these locations the average daily temperature never exceeds 0oC, degree day estimates of surface melting would not indicate melting in either station. We find that the S1-SAR finds 80 to 133 days of melting at Mt. Everest Camp II while the SEB indicates 100 days. At Mt. Everest South Col the S1-SAR finds either 19 or 77 days of melting while the SEB indicates 56. This analysis and discussion of the results will be added to the manuscript.

Based on comparisons with SEB modeling, and the physical basis of SAR measurements we have a high degree of confidence in our methodology and in the ability of the SAR backscatter to detect melting and in data-poor regions such as HMA remote sensing records of this type provide vital information supporting the understanding of surface melt on mountain glaciers. This is crucial to understand SEB on glacier and associated links to climate and downstream hydrological and ecological processes (Kayastha, Steiner et al. 2019).

**Figure AR1.** Air temperature measurements from (a) the Everest Camp II automated weather station (AWS) and (b) the Mt. Everest South Col AWS are compared to glacier

TCD
surface melting observations from the Sentinel-1 satellite synthetic aperture radar (SAR). (c) The radar backscatter from the Khumbu Glacier (at 6483 m a.s.l.) adjacent to the Camp II AWS, show a pronounced decrease in backscatter over several months associated with on-going surface melting during summer months. Melting is identified when backscatter decreases below a threshold (dashed-line), set at 3 dB below the winter mean (solid-line). (d) At the upper reaches of the Khumbu Glacier (7128 m a.s.l.), S1-SAR observes melting during ascending passes (18:00 local time) but not during descending passes (06:00 local time) except for a brief period during late June. (e, f) Timing of surface melt from observation and SEB modeling (Matthews et al., 2020) are compared to S1-Asc and Dsc observations at Camp II and South Col AWS, respectively. The cumulative number of melting days from the SEB model and S1-SAR (g, h)

... Even though in situ measurements might be absent, the authors might compare the melt signals retrieved from Sentinel-1 SAR data to other remote sensing sources (optical e.g. Landsat-8 and/or Sentinel-2 and SAR e.g. Radarsat-2). **AR4:** Current sources of thermal infrared measurements lack either the resolution (MODIS, AVHRR) or the temporal repeat (Landsat, Sentinel-2) to fully validate our surface melting product. Also, thermal infrared observations are somewhat limited by atmospheric effects (e.g. moisture, clouds), leading to uncertainties in determining the temperature at the surface under uncertain or indeterminate (unmeasured) weather conditions in HMA. Microwave sensors, e.g. SAR and radiometers are considered the best available option for detecting thaw and melt onset associated with the presence of liquid water (Entekhabi, Njoku et al. 2010). As stated previously, SAR backscatter responds to the liquid water content of the snow directly, rather than surface temperature. This backscatter response is pronounced and unambiguous. Although it would be interesting to explore the combined efficacy of thermal infrared and radar backscatter for characterization of mountain glaciers, thermal infrared observations do not provide a suitably robust standard for validating melt events characterized

**TCD**
by microwave SAR. A full comparative analysis is beyond the scope of this current manuscript and recommended for future research.

The four glacier regions studied are rather large, and it would have been interesting to understand more of the local variations of backscatter intensity values within these regions. A suggestion is to pick out some glaciers from each region and compare results from these individual glaciers.

**AR5:** We fully agree with the reviewer and have reconstructed the summary statistics by glacio-climatic region of HMA detailed in Bolch, et al (2019) (Bolch, Shea et al. 2019). Plots of z-score by elevation for each su-bregion. Please see the new figure below (Figure AR2).

**Figure AR2.** (Top) Glacio-climate subregions within the Hindu Kush Himalaya delineated in Bolch et al. (2019). (Middle) Mean z-score (2017-2019) by 100m SRTM elevation bin over each sub-region in the HKH. (Bottom) Mapped glacier area from the GAMDAM database (Sakai, 2019) over 100m SRTM (Farr 2007) elevation bins for each subregion.

**If possible, the authors should discuss the refreezing process in the percolation zone in the Hindu Kush Himalaya region in more detail.**

**AR6:** We agree that discussion of percolation zone backscatter signatures are limited and, with new analyses by glacio-climatic subregion used to summarize melt retrievals, we believe that the Results and Discussion section, along with subsection 4.1 on percolation zone meltwater and radar backscatter, could benefit from some restructuring. We propose a restructuring of the entirety of section 4 (Results and Discussion) and 4.1 (Percolation Meltwater Hydrology) as follows:
**Discussion**

A melting signal (z > 2) is observed across 97% (62,907 km2) of the mapped glacier area contained in the GAMDAM inventory. Melt retrievals are aggregated across 12 glacio-climate subregions within the HKH delineated by Bolch et al. (2019) and averaged across the calendar years 2017-2019 to report summary statistics. Aggregate statistics of melt onset (MO) and freeze onset (FO) are calculated across 100m elevation bins using the 30m SRTM (Farr 2007)digital elevation model for each glacioclimate subregion as presented in Figure AR3. For all subregions, there is a roughly linear relationship of mean MO with elevation over most elevations. The progression of MO with increasing elevation is consistent with lapse rate temperature controls on surface melting for most elevation ranges. Notably, we find an inflection toward earlier melt onset occurring at higher elevations (>6,500m a.s.l.). A divergence from lapse-rate driven melting at high elevations suggests that snowmelt onset may have regional triggers, like strong solar insolation (Matthews et al., 2019) or variable regional weather patterns: such as increases in atmospheric moisture, cloudiness, and deep convection induced by absorbing aerosols (Lau, Kim et al. 2010). In the three years of freeze onset (FO) across subregions we do not find the level of elevation dependence as observed in melt MO (Fig. AR3). For much of the HMA, FO occurs during a short period of time and over large spans of elevation. For example, in the Central Himalaya subregion, FO spans an average of 37 days while the MO for this region spans 78 days on average. Freeze onset (FO) across subregions does not follow a linear trend with elevation similar to melt MO (Fig. AR3). In western subregions (Eastern Hindu Kush, Western Pamir, and Karakoram), there is a signal of delayed refreeze apparent in summary statistics at higher elevation ranges within each respective catchment (Supplementary Fig. 1). In the Western Pamir, FO at 6,000m occurs 22 days later than FO at 5,000m a.s.l. Similarly, in the Karakoram, FO occurs 10 days later on average at 7,500m a.s.l. compared to 6,500m a.s.l. In the Tanggula Shan, FO at 6,500m a.s.l. is delayed by 21 days relative to FO at 5,500m a.s.l. (Supplementary Fig. AR1). Signals of delayed refreeze are observed at elevation ranges similar to greatest z-score across

[revised manuscript text omitted]

**AR8:** Glacier outlines were updated using data from the Landsat archive spanning 1990-2010. We suggest adding information on the sourcing of the glacier outlines by changing L 70 P 3 to read:

"The recently recently updated glacier outlines were derived from satellite optical imagery captured across the HKH by the Landsats 5 and 7 between 1990-2010 (Sakai, 2019)."

L 85, P 3: "In areas that are seasonally snow-free, like for areas of debris-cover or bare ice, melting conditions are dominated by surface scattering that is significantly darker relative to winter conditions (Lievens et al., 2019)." What is meant by "darker"? Is this connected thoroughness differences between debris-cover, bare ice and wet snowshoeing different backscatter intensity?

**AR 9:** We acknowledge that this phrasing is unclear and propose to change L 85, P3 as follows:

"In areas that are seasonally snow-free, like ablation zone debris-covered or bare ice surfaces, melting conditions are dominated by surface scattering once snowpack has completely melted away, which produces Sentinel-1 VH imagery with significantly lower backscatter intensity, causing SAR backscatter images to appear much darker over these zones when the glacier surface is no longer snow-covered relative to winter

TCD
conditions of deep and frozen snowpack (Lievens et al., 2019)"

**TCD**
L 95, P 4: "Limited only by the frequency of observations (12-days per orbit direction)". Are 6-days repeat orbits of Sentinel-1 available in the study region? If not, will 6-days repeat be available in the future? If so, this should be discussed. AR 10: We will add the following discussion following L95, P4:

Six-day repeat orbits are only available by combining ascending and descending paths. Because of the complexity of the terrain and varying look angles by orbit direction and relative orbit cycle number, we restricted SAR processing to distinct orbit directions and chose the orbit cycle and direction with greatest annual z-score in order to select an orbit direction and orbit cycle to choose for melt classification. If we attempted to combine orbit directions, our derived product would have inconsistent temporal resolution, as some orbit paths and directions pass the z-score criteria but in other regions only one orbit direction will capture the melt signal due to the complexity of the terrain. For the purposes of maintaining consistency throughout the derived dataset, we restricted to choosing only one orbit direction and one orbit pass within that direction to classify melt per pixel, which results in melt retrievals of 12-day resolution.

L 155-160, P 7: Why only Sentinel-1 SAR imagery from 2017-2019? Sentinel-1A was operative since 2014.

AR 11: We will add the following sentence after L154, P7:

Cross-polarized Sentinel-1 SAR did not become available over the HKH until early 2017 and thus restricted this study timeframe.

L 160, P 7: "By combining orbit directions, we utilize observations acquired at day and night. For the purpose of this study we do not attempt to resolve diurnal-scale melt freeze processes and instead focus on retrieving annual characteristics of melt timing and duration." The backscatter intensity signal differs between day and night. Further explanation on how this might affect the results must be included.

**AR 12:** The backscatter intensity signal will only vary significantly as refreeze processes progress within the snowpack and percolation areas. For the purposes of determining freeze/thaw status across a snowpack on a glacier surface, the retention of meltwater in the snow, and resulting snow wetness, is understood to persist until complete refreeze has taken place (Samimi and Marshall 2017). At C-band frequencies, this snow wetness will be visible across the volume of the snowpack (or firn strata) and diurnal refreeze processes that may be active at the surface of the glacier will not alter retrievals of wet snow across the volume.

L 261, P11: "Mean seasonal melt magnitude averaged over 100m elevation bins over all three calendar years of data shows strong (z > 2) melt signals across glacio-climatic sub-regions and across all elevation ranges of significant glaciation (Fig. 4). The occurrence of seasonal melt signals across all ranges of elevation in the HKH is both noteworthy and striking." These statements should be placed in the Results and Discussion section.

**AR 13:** These sentences will be moved accordingly.

L 315, P 14: "A summary of glacier melt timing with elevation averaged across study years is shown across 100m elevation bins in Figure 5 and tabulated across 1 km elevation bins in Table 2. Due to errors in melt classification at elevations below 3,500m a.s.l., we will summarize our observations across elevations >3,500m a.s.l. and assume that the relatively small glaciated area at

**TCD**
**elevations at or below 3,500m a.s.l. is negligible for the purposes of identifying trends across regions and elevation." What kind of errors are referred to? And What are the assumptions based on?**

**AR 14:** We hypothesize that erroneous melt classification at low elevations is due to combined effects of a shallow seasonal snowpack and heterogeneities across ablation zone surfaces (like debris-cover, sun-cups, supra-glacier meltwater, etc.), complicating radar backscatter returns to the sensor. Since less than 2% of glacier area exists at elevations less than 4,000m a.s.l. We believe that sparse results at the lowest glaciated elevations should not inhibit the interpretation of the dataset across regions with pronounced melt signals (z > 2).

**Technical comments**

**Figure 1: Top: Glacier outlines are not that apparent in the figure. Bottom: Blue color of descending swaths would be clearer.**

**AR 15:** To illustrate the number of glaciers within glacio-climatic subregions of the HKH, in Figure 1 we summarized the total glacier area by subregion and highlighted some of these glacier outlines with an inset map. We have also changed the colors of the descending swath paths to blue.

**Figure AR5.** (Top) Hindu Kush Himalaya (HKH) region and 2018 GAMDAM glacierized areas summed across glacio-climate subregions from Bolch, et al. (2019). An inset map highlights the spatial fidelity of GAMDAM outlines in the top panel. GGI and HKH data are overlaid onto a 30m 135Shuttle Radar Topography Mission (SRTM) DEM hillshade (Farr 2007). (Bottom) Sentinel-1 ascending (red) and descending (blue) swath footprints acquired across the study region. Ascending orbit cycle number 56 is highlighted in red to illustrate the SAR image processing approach for time series analysis across distinct orbit cycles.
Figure 2: Figures are too small and dark, and are hard to interpret. A suggestion is to cut down on the amount of figures and enlarge the most important ones that indicates VH polarization which is used in this work. Consider a brighter color on glacier outlines.

**AR 16:** We have cut down on area in these figures and highlighted the VH polarization, per your suggestion.

**Figure AR6.** (A) Mean summer (July-August) 2018 cross-polarized (VH) backscatter across an example region in the Trishuli basin, Nepal. (B) Mean 2018 winter (January-February) VH backscatter from Sentinel-1. (C) Sentinel-2 false-color (near-infrared, green, blue) image acquired by Sentinel-2 on October 30, 2018. Glacier outlines are shown in blue and the Yala glacier base camp meteorological station is marked in red. Note the snow covered and bare-ice portions of outlined glaciers and other debris-covered portions of glacier ablation areas. (C) The difference between mean summer and winter VH backscatter from Sentinel-1.

**Figure 3: Use same scale in the plots for VV and VH backscatter.**

**AR 17:** We have changed the plot to include both polarizations on the same color scaling.

**Figure AR7.** Time-series chart of air temperature measured at the Yala glacier base camp (4,950m a.s.l) andSentinel-1 A/B descending backscatter averaged across the Yala glacier for the years 2017-2018. Assessment of algorithm performance assuming mean daily air temperatures above 0°C indicates active melt results in 96% accuracy for melt classification across this time series in the VH polarized backscatter.
Figure 5: Suggestion to have elevation on the y-axis. Make the plots broader so more information can be interpreted. Include stippled lines for each 50 DOY in the figure to help the reader to interpret the results.

**AR18:** We have swapped the axes on these plots after re-running the analysis using glacio-climatic sub-regions delineated in Shean, et al. (2019) and have also created a supplementary figure (Supplementary Fig. AR1) to highlight melt retrievals and z-scores within each subregion.

**Figure AR8.** Melt onset (MO) and freeze onset (FO) summarized in 100m elevation bins using the 30m SRTM digital elevation model (Farr 2007) and 12 glacio-climate subregions (Bolch, 2019).

Figure 6: Difficult to understand and interpret the small circular insertions in the plot. Make them larger and explain more carefully, if these are of big importance.

**AR 19:** We split the inset maps into a separate figure: one to highlight the results of the melt retrieval algorithm presented in this work, and another to summarize melt retrievals across subregions delineated in Shean, et al. (2019).

**Figure AR9.** Melt retrievals averaged over the calendar years 2017-2019 in the Central Himalaya and Karakoram regions. (A) Mean melt onset (DOY) in the Khumbu glacier region of the Central Himalaya. (B) Mean melt onset (DOY) over the Siachen glacier in the Karakoram region. (C) Mean melt offset (DOY) in the Khumbu glacier region of the Central Himalaya. (D) Mean melt offset (DOY) over the Siachen glacier
in the Karakoram region.

**Figure AR10.** Melt onset (top) and freeze onset (bottom) averaged over 2017-2019 plotted over a SRTM 30m DEM hillshade (Farr 2007). Melt retrievals are averaged across glacio-climate subregions identified in Bolch, et al. (2019) and scaled by the mapped glacier area within each subregion.

Figure 8: Suggest to show the location on a map and include glacier outlines. AR 20: We have added an inset map to Figure 8.

References should be in chronological order. e.g. L72, P

**AR 21:** We will restructure references to make sure they are in chronological order.

**References for Author Response 1**

Ashcraft, I. S. and D. G. Long (2001). Azimuth variation in microwave backscatter over the Greenland ice sheet. IGARSS 2001. Scanning the Present and Resolving the Future.

Proceedings. IEEE 2001 International Geoscience and Remote Sensing Symposium (Cat. No. 01CH37217), IEEE.

Ashcraft, I. S. and D. G. Long (2007). "Comparison of methods for melt detection over Greenland using active and passive microwave measurements." International Journal of Remote Sensing 27(12): 2469-2488.
Bolch, T., J. M. Shea, S. Liu, F. M. Azam, Y. Gao, S. Gruber, W. W. Immerzeel, A. Kulkarni, H. Li and A. A. Tahir (2019). Status and change of the cryosphere in the Extended Hindu Kush Himalaya Region. The Hindu Kush Himalaya Assessment, Springer: 209-255.

Brangers, I., H. Lievens, C. Miège, M. Demuzere, L. Brucker and G. De Lannoy "Sentinel-1 detects firn aquifers in the Greenland Ice Sheet." Geophysical Research Letters.

Entekhabi, D., E. G. Njoku, P. E. O'Neill, K. H. Kellogg, W. T. Crow, W. N. Edelstein, J. K. Entin, S. D. Goodman, T. J. Jackson and J. Johnson (2010). "The soil moisture active passive (SMAP) mission." Proceedings of the IEEE 98(5): 704-716.

Farr, T. G., Rosen, P.A., Caro, E., Crippen, R., Duren, R., Hensley, S., Kobrick, M., Paller, M., Rodriguez, E., Roth, L., Seal, D., Shaffer, S., Shimada, J., Umland, J., Werner, M., Oskin, M., Burbank, D., and Alsdorf, D.E. (2007). "The shuttle radar topography mission: Reviews of Geophysics, v. 45, no. 2, RG2004, at https://doi.org/10.1029/2005RG000183." Reviews of Geophysics 45(RG2004).

Fischer, G., M. Jäger, K. P. Papathanassiou and I. Hajnsek (2019). "Modeling the Vertical Backscattering Distribution in the Percolation Zone of the Greenland Ice Sheet with SAR Tomography." IEEE Journal of Selected Topics in Applied Earth Observations and Remote Sensing 12(11): 4389-4405.

Google. (2020). "Sentinel-1 Preprocessing." Retrieved Novermber 30, 2020, 2020, from Sentinel-1 Preprocessing.

**TCD**
Gorelick, N., M. Hancher, M. Dixon, S. Ilyushchenko, D. Thau and R. J. R. S. o. E. Moore (2017). "Google Earth Engine: Planetary-scale geospatial analysis for everyone." 202: 18-27.

Jezek, K. C., P. Gogineni and M. Shanableh (1994). "Radar measurements of melt zones on the Greenland ice sheet." Geophysical Research Letters 21(1): 33-36.

Johnson, A., M. Fahnestock and R. Hock (2020). "Evaluation of passive microwave melt detection methods on Antarctic Peninsula ice shelves using time series of Sentinel-1 SAR." Remote Sensing of Environment 250: 112044.

Kayastha, R. B., N. Steiner, R. Kayastha, S. K. Mishra and K. McDonald (2019). "Comparative study of hydrology and icemelt in three Nepal river basins using the glacio-hydrological degree-day model (GDM) and observations from the Advance Scatterometer (ASCAT)." FrEaS 7: 354.

Lau, W. K., M.-K. Kim, K.-M. Kim and W.-S. Lee (2010). "Enhanced surface warming and accelerated snow melt in the Himalayas and Tibetan Plateau induced by absorbing aerosols." Environmental Research Letters 5(2): 025204.

Margulis, S. A., Y. Liu and E. Baldo (2019). "A joint Landsat-and MODIS-based reanalysis approach for midlatitude montane seasonal snow characterization." Frontiers in Earth Science 7: 272.

Matthews, T., B. Perry, D. Aryal, D. Shrestha and A. Khadka (2019). New Heights in Glacier-Climate Research: Initial Insights From the Highest Weather Stations on

**TCD**
Earth. AGU Fall Meeting 2019, AGU.

Matthews, T., L. B. Perry, I. Koch, D. Aryal, A. Khadka, D. Shrestha, K. Abernathy, A. Elmore, A. Seimon and A. Tait (2020). "Going to Extremes: Installing the World's Highest Weather Stations on Mount Everest." Bulletin of the American Meteorological Society. Paterson, W. S. B. (2016). The physics of glaciers, Elsevier.

Samimi, S. and S. J. Marshall (2017). "Diurnal cycles of meltwater percolation, refreezing, and drainage in the supraglacial snowpack of Haig glacier, Canadian Rocky Mountains." Frontiers in Earth Science 5: 6.

Shi, J. and J. Dozier (1995). "Inferring snow wetness using C-band data from SIR-C's polarimetric synthetic aperture radar." IEEE transactions on geoscience and remote sensing 33(4): 905-914.

Steiner, N. and M. Tedesco (2014). "A wavelet melt detection algorithm applied to enhanced-resolution scatterometer data over Antarctica (2000–2009)." The Cryosphere 8(1): 25-40.

Trusel, L. D., K. E. Frey, S. B. Das, P. K. Munneke and M. R. Van Den Broeke (2013). "Satellite-based estimates of Antarctic surface meltwater fluxes." Geophysical Research Letters 40(23): 6148-6153.

Ulaby, F. T. and W. H. Stiles (1980). "The active and passive microwave response to snow parameters: 2. Water equivalent of dry snow." Journal of Geophysical Research: Oceans 85(C2): 1045-1049.
**Interactive comment on The Cryosphere Discuss., https://doi.org/10.5194/tc-2020-181, 2020.**

---

## Author Comment (AC2) · 30 Nov 2020

**Author Response to Reviewers 2**

This paper provides an application of C-Band Sentinel-1 imagery to map seasonal melting along the Hindu Kush Himalaya (HKH) for the period 2017-2019. The paper provides spatiotemporal time series of several metrics, as for example freeze onset, for the region. The authors detected horizontal (at the regional scale) and vertical differences that they assess resemble well established HKH glacio-climatic regimes while providing insights on scarcely described dynamics, such as the occurrence of melt signals at very high elevations. My

**general assessment is that the paper is straight: it describes a method that naturally connects with the results. The application by itself is of great value in other regions for monitoring as well as for model validation, given the high resolution of the imagery and the mission's expected length of 7 years. After reading the paper several times, however, I am not fully convinced the paper fits in this journal in its present form instead of a more remote sensing oriented journal. I think it lacks a more in depth glacier-climate interpretation. Below I include some comments that argue my assessment.**

**General Response**

We thank the Referee for their close consideration of our work. We have completed major revisions to the manuscript to aggregate melt retrievals by clearly articulated glacio-climate regions from Bolch, et al. (2019) and investigate high elevation melting signals with surface energy balance models of glacier melt and in situ automatic weather station (AWS) data from Matthews, et al. (2020) (Bolch, Shea et al. 2019, Matthews, Perry et al. 2020).

We have re-constructed summary statistics of melt retrievals across glacio-climatic sub-regions delineated in Bolch, et al. (2019) in order to create a baseline measurement of what will hopefully become a climatic record over the mission lifespan of Sentinel-1. We went through each figure in the manuscript and replotted data based on the delineations of sub-regions characterized by Bolch et al. (2019). We have included all of these figures in our author response (AR) (please see Author Response to Referee 1) and believe that summary statistics by recently defined glacio-climate subregions helps to illustrate that Sentinel-1 melt retrievals are able to capture heterogeneities in melt characteristics across glaciers and glacio-climate subregions conducive to support glacio- and meteo-climate studies across the HMA. We contend that Sentinel-1 snowmelt retrievals across a three-year record provide an important baseline measurement of melting within regions of High Mountain Asia (HMA). We have modified the discussion accordingly, and propose a brief section

comparing relative melt durations with post-2000 mass balance characteristics across HMA sub-regions tabulated in Bolch, et al (2019).

We hope that the development of an observational dataset on the glacier surface energy balance will provide means of model calibration and validation; supporting research into HMA weather, climate, and hydrology. Much of the remote sensing methodology used in our approach is very well established in prior research. It is applied here to address open questions in cryospheric sciences, for example, constraining the glacier surface energy balance. We provide detailed responses below.

**General comments**

**(1) The paper provides a detailed description of the methods but to me the section "Results and Discussion" lacks discussion on glacier-climate regimes. The authors claim that melt dynamics coincide with different HKH glacio-climatic regimes. As far as I can tell, these regions correspond to operational inventory subdivisions (for CGI or RGI) and do not necessarily correspond to boundaries between well-defined glacier regimes. Perhaps just rearranging the narrative will clarify this, but I would suggest to expand the discussion including some reasons why these areas show these differences or whether other regional differentiation is possible.**

**AR 22:** We have undertaken a re-analysis of our results to summarize by glacio-climatic subregions from Bolch, et al. (2019). For the purposes of elucidating differences between glacio-climatic regions of the HMA, we re-plotted all figures to resemble regions delineated in Bolch et al. (2019) and have included them in Figures AR1, AR2, AR3, and AR4. We have similarly proposed modifying the "Results and Discussion" section to incorporate comments and concerns from yourself and Referee 1. The re-worked "Discussion and Conclusion" section is detailed in AR6 to Referee 1. We suggest appending the "Discussion and Conclusion" section detailed in AR6 with a

brief discussion on glacio-climate regions, mass wasting, and Sentinel melt retrievals, by appending the following section to our discussion:

**Melt Retrievals and Glacio-Climate Regimes**
The three-year record of Sentinel-1 SAR retrievals of glacier melt status represent a baseline measurement for the HMA. The summary melt statistics are aggregated over glacio-climate subregions in a comparison between melt retrievals and subregional estimates of glacier mass wasting (Bolch, et al. 2019). Overall, the HMA subregions with the most rapid mass wasting between 2000-2010 tabulated in Bolch, et al. (2019) (Eastern Himalaya, Hengduan Shan, Nyainquntanglha) exhibit the greatest number of melt days on average in 2017-2019 from Sentinel-1 melt retrievals. Subregions with slower mass wasting, in steady state, or with mass slight gain (Eastern Hindu Kush, Western Pamir, Karakoram, Tibetan Interior) show on average one month less of melt duration relative to regions with accelerated mass wasting. Interestingly, the Gangdise and Eastern Tibetan Mountains subregions, both with some of the higher post-2000 rates of glacier wasting in the HMA, show annual melt durations of roughly 3 and 4 months respectively, which appears more characteristic of western regions of slower mass wasting. Although Sentinel-1 retrievals of glacier melt status for three calendar years does not make-up a climatic record, we observe that between 2017-2019 there was on average less duration of melting in regions where in situ data and climate models indicate that frozen winter precipitation contributes to glacier accumulation despite warming global climate (Karakoram, Hindu Kush, Eastern Pamir, Western Himalaya) (Palazzi, Von Hardenberg et al. 2013, Kapnick, Delworth et al. 2014, Kääb, Treichler et al. 2015). We interpret shorter duration of annual melt days in the western regions of the HMA as a potential indicator of the "Karakoram Anomaly" reflected in the Sentinel-1 data record. Because the meteoclimatic drivers of the Karakoram Anomaly are still under debate (Farinotti, Immerzeel et al. 2020), Sentinel-1 retrievals of melt duration might be useful for interrogating meteoclimatic drivers of heterogeneity in glacier wasting dynamics across the HMA.

**(2) Along the same lines, I believe the paper needs to add some sort of longer climatic discussion. Currently, lines 418-419 present a reworked version of 131-132. Is it possible for the authors to include some literature or analysis on the possible weather patterns that would explain the findings for the studied period? Perhaps looking at re-analysis fields for the MO and/or FO days will allow the authors to further elaborate on the findings. In addition, this can be helpful in trying to explain the high elevation melting events.**

**AR 23:** We have investigated high elevation melting events using in situ data from Matthews, et al. (2020). After analyzing surface energy balance (SEB) results and in situ AWS stations (detailed in AR3), we have added discussion on the relevance of surface air temperature to the retrieval of the presence of liquid water within a snow or firn matrix. We propose that, for the scope of this paper, we present relevant observational data and a methodology for operationalizing retrieval of these data, alongside an example of how these data can be used to constrain SEB model results from in situ AWS data. We seek to support further research into the climate and weather patterns governing glacier mass balance in the HKH but feel that broadening this discussion in too much detail is beyond the scope of the results that we are able to provide; including reanalysis temperature data, as it is likely to poorly constrain or describe melt dynamics at due to the SEB controls active at elevations greater than the 0°C isotherm.

**(3) In line 374 I'm not sure if longwave energy is what drives melt at these elevations. I think that the Everest climate data is showing that the large input of shortwave energy and monsoonal activity allow for the glacier surface to reach melting temperatures despite below freezing air temperatures. Since the authors are seen this possible effect, I think there is a great opportunity to test whether some of the high-elevation melt events coincide with the conditions depicted in Matthews et al (2020), by looking at weather conditions around those**

[Figure]

**dates.**

**AR 24:** We agree, the language in this section is misleading. We do not contend that longwave energy alone drives melting at high elevations. To better understand high elevation melting we have analyzed data from Matthews, et al (2020) in terms of air temperature and SEB characterizations of glacier melt. A new figure is included alongside an accompanying discussion as detailed in AR3 to Referee 1.

**(4) L184: Can you provide more details on the methods included in the computing**

**AR 25:** We will broaden our discussion on the computing method as detailed in AR1 to Referee 1.

**(5) L206: Is it possible to attempt a sensitivity analysis of the b value? Could the choice of the b value be partially contributing to uncertainty in the results?**

**AR 26:** The b value is a commonly used threshold to separate dry snow from melting snow. Since we lack widespread in situ measurements of surface melting, it is difficult to determine the optimal threshold in a systematic way. Compared to previous studies, 3 dB is the most common and most conservative of thresholds used in previous studies. To the best of our knowledge all previous studies have chosen a threshold between 1 and 3 dB.

**(6) Figure 3: I think the validation data is insufficient to make general statements on the method for the whole region. I wonder if it is possible to make a comparison relative to reanalysis air temperature at equivalent geopotential levels for a larger region.**

**AR 27:** It would be interesting to compare air temperatures from reanalysis fields however, we do not think these are appropriate to use in a validation effort. First, there is a large resolution mismatch between available reanalysis data and S1-SAR

observations. Further, downscaling estimates of snowmelt are prone to large errors especially in topographically complex terrain (Baldo and Margulis 2017). There is also a well-known cold bias in most reanalysis products (Margulis, Liu et al. 2019). Lastly, the basis of our observational record is the radar response to liquid water, and the radar has no direct temperature dependence. We show in Figure AR1 to Referee 1 that backscatter can be a more accurate indicator of melting than temperature alone. As discussed in our response to Reviewer 1, to address measurement uncertainty to the best of our ability we have added a comparison of in situ data as presented in Matthews et al (2020) and detailed in AR3.

**(7) Figure 5: wouldn't it be more straightforward to leave elevations in the y axis and DOY in the x axis? Also, although the spread of DOY do not seem to be statistically different there is an interesting divergence between about 5700 to 6700, where Karakoram and Western Himalaya tend to cluster differently relative to Central and Eastern Himalaya. Perhaps studying some air temperature and humidity dynamics at the corresponding geopotential levels will allow the authors to interpret that situation.**
**AR 28:** We have switched the axes on Figure 5 and also recalculated melt statistics using glacio-climate regions delineated in Bolch, et al. (2019). Melt retrievals show interesting separation when calculated over these subregions. We have proposed to add analysis and discussion on air temperature and SEB dynamics as detailed in AR3.

**References for Author Response to Referee 2**

Baldo, E. and S. A. Margulis (2017). "Implementation of a physiographic complex­ity‐based multiresolution snow modeling scheme." Water Resources Research 53(5): 3680-3694.
Bolch, T., J. M. Shea, S. Liu, F. M. Azam, Y. Gao, S. Gruber, W. W. Immerzeel, A. Kulkarni, H. Li and A. A. Tahir (2019). Status and change of the cryosphere in the Extended Hindu Kush Himalaya Region. The Hindu Kush Himalaya Assessment, Springer: 209-255.

Farinotti, D., W. W. Immerzeel, R. J. de Kok, D. J. Quincey and A. Dehecq (2020). "Manifestations and mechanisms of the Karakoram glacier Anomaly." Nature Geoscience 13(1): 8-16.

Kääb, A., D. Treichler, C. Nuth and E. Berthier (2015). "Brief Communication: Contending estimates of 2003–2008 glacier mass balance over the Pamir–Karakoram–Himalaya." Cryosphere 9(2).

Kapnick, S. B., T. L. Delworth, M. Ashfaq, S. Malyshev and P. C. Milly (2014). "Snowfall less sensitive to warming in Karakoram than in Himalayas due to a unique seasonal cycle." Nature Geoscience 7(11): 834.

Margulis, S. A., Y. Liu and E. Baldo (2019). "A joint Landsat-and MODIS-based re-analysis approach for midlatitude montane seasonal snow characterization." Frontiers in Earth Science 7: 272.

Matthews, T., L. B. Perry, I. Koch, D. Aryal, A. Khadka, D. Shrestha, K. Abernathy, A. Elmore, A. Seimon and A. Tait (2020). "Going to Extremes: Installing the World's Highest Weather Stations on Mount Everest." Bulletin of the American Meteorological Society.

Palazzi, E., J. Von Hardenberg and A. Provenzale (2013). "Precipitation in the Hindu‐Kush Karakoram Himalaya: observations and future scenarios." Journal of Geophysical Research: Atmospheres 118(1): 85-100.
* * *
Interactive
comment

---

## Author Response (AR1)

Dear Dr. Sandells,

Many thanks for your thoughtful reviews. We have updated our manuscript with the major revisions you have suggested. Please find our detailed responses below.

With best wishes,

Nick Steiner

**..include more detail on the methodology and broaden the manuscript into energy balance modelling..**

We have included 2 new sections describing surface energy balance modelling inputs (Section 2.4), and results and methodology (Section 3.3). We have included discussions these results in the abstract, discussion and conclusions. In addition, some detail on the SEB parameterization is included as in a Supplementary document.

**... include ... sub-region analysis ...**

We have included analyses of melt retrievals by glacio-climate sub-regions delineated by the Hindu Kush Himalaya Monitoring and Assessment Program (HiMAP) to aggregate Sentinel-1 melt retrievals across sub-regional delineations that are most relevant to current literature.

**Please could you ensure the correct Figure AR7 has been included (at the moment AR7 is a duplicate of Figure 8).**

We have ensured that all figures are correctly cited and captioned.

**AR3 is also the same as AR8 but has the same caption.**

We have removed any duplicate figures that appear in the author's response

**Please could you also revise Figure AR2 as similar coloured lines are difficult to distinguish in AR2b and AR2c (perhaps use different line styles).**

We have changed the color palette and differentiated subregions using multiple line-styles.

---

## Referee Report (RR1)

**Review of "Mapping seasonal glacier melt across the Hindu Kush Himalaya with time series SAR" by Scher et al.**

This study uses Sentinel-1 SAR data to map the seasonal melt characteristics (melt onset, freeze onset, and melt duration) across the Hindu Kush Himalayas.  Using this dataset, the study investigates spatio-temporal variations across subregions and as a function of elevation.  Two sets of automatic weather stations are used to validate the interpretation of these studies and highlight where glacier ablation models would perform poorly.  Given the lack of in-situ observations and seasonal remotely sensed observations in this region, this type of dataset would be very valuable to modeling glacier melt.  The use of systematic C-Band SAR observations in this region also appears to be the first time this has been done now that the satellite imagery is available.

While I believe this would be a valuable contribution to the field, I unfortunately believe there to be several major issues with the interpretation of the dataset.  The primary issue being the interpretations of the ablation area, which the authors note is challenging and/or were excluded; although this exclusion is unclear because analyses suggest that all area were included.  Furthermore, for a study of this scale, I expected there to be some type of validation to support their findings.  Instead, the two sets of automatic weather stations were partially used for validation and partially used to highlight where melt models would perform poorly.  I read the author's response to reviewer's where they state that this is not possible, except for the two automatic weather stations; however, I don't agree with this assessment.  For example, other relevant studies (some pointed out in the review below), in-situ measurements, and/or other sources of satellite imagery (other microwave datasets, optical datasets, etc.) would have been useful to support the interpretations and conclusions.

In my opinion, there needs to be considerably more validation.  I would also suggest that unless the methods can be improved to handle the challenges in the ablation area (which would need to be shown through a rigorous validation process to provide confidence), that the ablation area be excluded entirely, and the results be limited solely to the timing of melt in the accumulation zones.  Another alternative is that if debris-covered areas are the only problem, then limit the study to clean-ice glaciers only.  Either way would still provide useful information to modelers, although this would greatly reduce the novelty and impact of the study overall.  At that point, the novelty of the study would need to be reconsidered.  I therefore recommend the paper to be reconsidered after major revisions.  Please see my major comments and minor comments below.

Major Comments
The methods show results (e.g., Figure 4 middle; Figure 5).  If these datasets were used to develop the method or solely for validation, then this should be explicitly stated.  Either way, the assessments of the methods should be in a separate results section.  This performance assessment could then focus on validating the methods before the large-scale assessment of spatial trends.

I found the references to support various statements and relevant work needs improvement. I have highlighted many examples in the introduction, but also believe this is necessary for validating the interpretation of the SAR signals as well.

The method does not appear to work over ablation areas. It is unclear from the results presented if this is an isolated issue with debris-covered areas or if it is an issue with clean ice as well because the example figures were only for debris-covered glaciers. Given debris-covered areas are highly prevalent in HKH, this is a major issue. This is clearly apparent from Figure 7 Top. The Khumbu Glacier is debris-covered below the Khumbu icefall. The interpretation of the SAR signal on this debris-covered area is that it indicates refreeze onset between DOY 250-270, which is in September. This is highly inaccurate and highlights the lack of validation within the study. For example, this part of the debris-covered ice is clearly still melting in September (Rowan et al. 2020; *Journal of Glaciology*). Similar issues appear to persist with the Freeze Onset in Figure 8C,D.

It's unclear if similar issues exist for the melt onset signals as well. The color bar in Figure 8A,B is too hard to read to discern if melt onset appears to be happening as early as March 1st, which would be unlikely. However, the fact that the color bar is shows DOY 60 suggests there are some areas where it is melting at this time; otherwise, why stretch the color bar outside the values shown in the figure? The issues with the freeze onset and melt onset and lack of any validation, in my opinion, undermine the entire study.

One recommendation is to put the results and interpretation into better context of glacier zones. For example, the discussion/interpretation primarily focused on the liquid water in frozen snow and percolation faces, but the results that were being interpreted were aggregated from 3000-7500 m.a.s.l. The lower elevations are clearly in the ablation zone, i.e., clean ice or debris-covered ice. Therefore, it's unclear how much of the discussion of the snowpack and percolation faces is warranted. Or is this partially being included to discuss seasonal snowpack covering the ablation zone? Including a rough estimate of the glacier zones, even if they are roughly estimated based on median elevation or some other metric, and their respective hypsometries may provide some useful context for interpreting the results. Otherwise, results/discussion like L406-414 come across as interpreting the entire glacier as being in the percolation zone.

Given the major issues with the ablation area, I would suggest either (i) limiting the study to only accumulation areas, or (ii) limiting the study to only clean-ice glaciers. Either way, there needs to be significantly more validation performed to provide confidence in the results. This validation should ideally be done for both the ablation and accumulation areas, albeit that the accumulation zones interpretation should be much stronger as they are based in theory as the authors clearly discuss in the main text and mention in response to a previous reviewer.

This validation is important as it is unclear how sensitive the results are to various aspects of the methods. For example, is there any sensitivity to the chosen dB threshold of 3? A previous reviewer asked for a sensitivity analysis, but the author's simply responded that this was

conservative based on literature values.  Even if this is only done for a handful of glaciers, a sensitivity analysis would provide more confidence.  Similarly, what about using ascending vs. descending orbits?  The choice of orbit direction clearly affects the interpretation of the signal at high elevations (Figure 5) and yet for the full region a composite was used.  If this composite is used moving forward, it'd be good to see some sensitivity/error analyses because it appears to have a strong impact on the results, but perhaps this is only for high altitudes.

Similarly, were the high-altitude automatic weather stations used for validation?  The energy balance modeling likely has its own issues (e.g., sublimation could be important here?), but the modeling appears to be used to validate the SAR signal (L320).  If it is used for validation, does that mean that the SAR signal overestimates the amount of melt by 33-43% (L345)?  This would appear to be a considerable amount of error.

Specific Comments
Given there's no word limit, I would suggest writing out acronyms like melt onset and freeze onset to make the study more readable to the average reader.

L34: "Ice caps" have a particular meaning.  In HKH, they are primarily (if not all) mountain glaciers or valley glaciers.

L29-46: This introduction bounces back and forth between discussing global issues (e.g., contributing 25% of sea level rise) to HKH specific issues (e.g., freshwater for 2 billion people). It's useful to show how HKH changes fit into the larger picture, but I'd be conscious of this change in scales to make it easier to read.

L47-49: Litt et al. (2019) focuses on two glaciers over a period of a several years.  It does not address "projected disturbances".  More appropriate reference would be one of the GlacierMIP studies (e.g., Hock et al. 2019 (*Journal of Glaciology*) or Marzeion et al. 2020 (*Earth's Future*)). The same is true for L55, which discusses projections but does not mention relevant studies where these uncertainties exist.

L65-68: Statement concerns inability of studies to capture variability in patterns and magnitude of melting, but only references a study that measures the geodetic mass change (Brun et al. 2017).  The more recent study by Shean et al. (2020; *Frontiers in Earth Science*) would be appropriate to reference here as it provides refined estimates of mass change compared to Brun et al. (2017).  Furthermore, the recent study on projections in this region (Rounce et al. 2020; *Frontiers in Earth Science*) explicitly captures the variability reported by these measurements using a degree-day model.

L73: "operational monitor" – consider changing "operational monitoring system" perhaps?

L127 – this would be useful for assessing all models of glacier ablation, not just energy balance models.

L137 – for *The Cryosphere* I would suggest using more standard glacier mass change terms (e.g., mass loss, mass balance, mass change) as opposed to the term "wasting" that is less frequently used.

L139-141 – appears to imply that Karakoram, Kunlun Shan, etc. are not affected by increases in global average temperature. I'd suggest references Karakoram anomaly and recent studies finding that they are starting to lose mass (e.g., Farinotti et al. 2020 (*Nature Geoscience*)).

L148 – is there a reference for this ICIMOD dataset?

L158-171 – this is a lot of detail on the glacier outlines used in a study that I don't see being relevant. If debris-covered areas are an issue, there's no mention of datasets that explicitly delineate it (Scherler et al. 2018 (*Geophyiscal Research Letters*); Herreid and Pellicciotti 2020 (*Nature Geoscience*)). This should be discussed.

L189 – "this" should be "the" or "the timeframe of this study"

L204 – suggest stating the name of the cloud-computing platform here, i.e., Google Earth Engine, to make this explicit for readers as opposed to only mentioning it at the end of the paragraph.

L216 – It is my understanding that these are both on Khumbu Glacier. Therefore, this should be "over a high elevation glacier", not glaciers.

L245 – Why is this weather station not stated in Section 2.4?

L247 – Introduction stated that assuming 0 degC threshold is poor assumption and was part of the motivation for this study. May want to preface that this generally works well, except for at extreme altitudes where other processes (e.g., sublimation) may be important (if that is the point that is trying to be stated)?

L296 – It's unclear how debris-covered areas were identified (see previous comment). Nonetheless, not including debris meant between 30-48% of the ice in the ablation area is not being monitored (Kraaijenbrink et al. 2017; *Nature*). It would be good to state how much over the actual glacier area was able to be monitored, and if statements are being made about the ablation area (like that being mentioned here), then stating the percentage of the area in the ablation area being monitored is relevant as well.

L307 – if the debris-covered area is excluded, then how are statements being made concerning those areas? Is the assumption at the regional scale that when evaluating variability, the debris-free areas that were measured at lower elevations are representative of the debris-covered areas? If so, this should be stated explicitly, as it's confusing within its present form. Note that the Khumbu Glacier example clearly shows debris-covered areas being included.

L322 – "are" should be "is"

L324 – If the energy balance modeling is an important part of this study, which it appears to be, then additional detail should be provided in the main text.  At a minimum, this should include the values used to force the model (i.e., those in Table SI) as well as relevant information pertaining to the timestep and the terms considered (i.e., Equation 1 in the supplement).

L330-350 – these are results, not methods.

L337-339 – I was thinking this same point, so I'm glad the authors brought this up.  However, the timing of satellite overpasses is known (Figure 5 caption), so why was this not performed?  This appears to be the primary validation of how well SAR performs, and one of the primary conclusions of the study that SAR can detect melting where models otherwise wouldn't, so it should be presented in a rigorous analysis to provide confidence.

L351-355 – High degree of confidence from two weather stations, where the days of melting appears to be overestimated by 33-43% (L345) seems to oversell the results.  I agree that this is a challenging topic, especially since there are uncertainties with the SAR data and uncertainties with the energy balance model (unfortunately, there's no validation data for the energy balance model, so are these differences due to the SAR data or the energy balance model – likely both?).

L355 – "HMA" (High Mountain Asia) is not defined in text.  It's used later in text as well.  I'd suggest using HKH throughout.

L386 – except for debris-covered areas, no (L297)?

Table 3 – caption and table are opposite directions making it very difficult to read.  Also, 1 km elevation bins are very coarse considering that some regions (if not most regions) have most of their glacier area spanning 1-2 km (Figure 4 Bottom).  This means that the statistics shown for the other elevation bins is for a tiny fraction of the glacier area, which in my opinion detracts from the overall value of this table.  The 100 m bins in Figure 4 are much more meaningful.

Figure 6 – this figure is difficult to read/interpret due to the lines.  I would suggest changing both the circles and the squares to lines.  The circles and squares are clearly separated, so you could simply add a note that the melt onset is on the left and freeze onset on the right, which is intuitive anyways.  This would enable the reader to follow the lines and determine how the trends vary.  It will also make it easier to see the 12 subregions, which currently are on top of one another and hard to discern any information from.  This will also make it easier to see statements like L400 where it does not follow a linear trend, when the current form of Figure 6 looks like roughly speaking higher elevations are around DOY 250-270 and lower elevations are around 280-330, so while the linear trend is not as strong, there still appears to be a trend.

L393-396 – is this supported by the energy balance modeling?

L429-433 – Khumbu Glacier is highly debris-covered below the Khumbu icefall, which the authors appear to interpret as a signal of refreeze onset.  This is highly inaccurate.  Figure 7 (Top) suggests the Khumbu Glacier would be refreezing on DOY 250-270 (sometime in September) when the debris-covered ice is clearly still melting (Rowan et al. 2020; *Journal of Glaciology*).  Figure 8C,D show similar issues.  See major comment.

L533 – Shean et al. (2019a) does not exist.  Unclear therefore where the trends for 2000-2010 are coming from and why trends in later decades covering the time period observed (i.e., 2000-2018 from Shean et al. 2020) are not used.

L557 – surface energy balance models "are constrained" by SAR data indicates that the SAR data is used to calibrate the energy balance models.  This is in direct contrast to earlier where the energy balance models were being used as validation of the SAR observations (L320).

---

## Author Response (AR2)

Dear Dr Sandells,

Thank you for your thoughtful review of our manuscript. To address your comments from we have changes as detailed below.

Best wishes,

Nick Steiner

Page 4, line 4: retirevals -> retrievals

Pg4Ln111 – We have corrected the spelling error.

Page 12, second paragraph: 'never above zero but experience above zero maximum glacier surface temperatures during starting in June 2019' should this be maximum air temperatures at the glacier surface?

Pg12Ln330 - Thank you for your comment. Here, we are referring to the maximum daily glacier surface temperature as calculated from SEB.  To clarify these results, we have corrected the text to (1) only discuss maximum daily temperatures and (2) indicate the SEB glacier surface temperature as $T_s$ and the daily maximum air temperature as $T_a$ as defined in the previous paragraph.

Page 17, 1st paragraph: '~5,400-6,2000m' -> ~5,400-6,200m

Pg17ln432 – Thank you for your comment, we have corrected the upper elevation range.

Page 18: 'brightening of the radar signal due to surface scattering contributions from wet debris'. Would absorption in wet debris result in a decrease in backscatter or is this potential effect more to do with roughness effects or the composition of the debris?

Pg18ln470 – Thank you for your comment, we agree that this statement needs to be clarified. We have added text to indicate that our understanding is that over ablation surfaces, like debris-cover, changes in backscatter during snow-off conditions may be dominated by scattering from surface roughness rather than absorption effects.

Supplementary material - is the longwave radiation measured or modelled?

The longwave flux is computed using measurements of incoming longwave radiation and the outgoing longwave radiation computed using the Stefan-Boltzmann law. We have updated the supplementary material to include this information.

---

## Author Response (AR3)

**Review of "Mapping seasonal glacier melt across the Hindu Kush Himalaya with time series SAR" by Scher et al.**

*This study uses Sentinel-1 SAR data to map the seasonal melt characteristics (melt onset, freeze onset, and melt duration) across the Hindu Kush Himalayas. Using this dataset, the study investigates spatio-temporal variations across subregions and as a function of elevation. Two sets of automatic weather stations are used to validate the interpretation of these studies and highlight where glacier ablation models would perform poorly. Given the lack of in-situ observations and seasonal remotely sensed observations in this region, this type of dataset would be very valuable to modeling glacier melt. The use of systematic C-Band SAR observations in this region also appears to be the first time this has been done now that the satellite imagery is available.*

*While I believe this would be a valuable contribution to the field, I unfortunately believe there to be several major issues with the interpretation of the dataset. The primary issue being the interpretations of the ablation area, which the authors note is challenging and/or were excluded; although this exclusion is unclear because analyses suggest that all area were included. Furthermore, for a study of this scale, I expected there to be some type of validation to support their findings. Instead, the two sets of automatic weather stations were partially used for validation and partially used to highlight where melt models would perform poorly. I read the author's response to reviewer's where they state that this is not possible, except for the two automatic weather stations; however, I don't agree with this assessment. For example, other relevant studies (some pointed out in the review below), in-situ measurements, and/or other sources of satellite imagery (other microwave datasets, optical datasets, etc.) would have been useful to support the interpretations and conclusions.*

*In my opinion, there needs to be considerably more validation. I would also suggest that unless the methods can be improved to handle the challenges in the ablation area (which would need to be shown through a rigorous validation process to provide confidence), that the ablation area be excluded entirely, and the results be limited solely to the timing of melt in the accumulation zones. Another alternative is that if debris-covered areas are the only problem, then limit the study to clean-ice glaciers only. Either way would still provide useful information to modelers, although this would greatly reduce the novelty and impact of the study overall. At that point, the novelty of the study would need to be reconsidered. I therefore recommend the paper to be reconsidered after major revisions. Please see my major comments and minor comments below.*

*Major Comments:*

*The methods show results (e.g., Figure 4 middle; Figure 5). If these datasets were used to develop the method or solely for validation, then this should be explicitly stated. Either way, the assessments of the methods should be in a separate results section. This performance assessment could then focus on validating the methods before the large-scale assessment of spatial trends.*

The z-score metric, Figure 4 middle panel, is indicative of the magnitude of the radiometric

response that we are isolating. This information is key when defining our methodological approach. Similarly, Figure 5 is used to confirm our interpretation of the radar signature in our methodology. We use the results section to communicate the major outcomes of the paper, a description of regional timings and extent of glacier surface melting.

*I found the references to support various statements and relevant work needs improvement. I have highlighted many examples in the introduction, but also believe this is necessary for validating the interpretation of the SAR signals as well.*

Thank you for your comment. We have addressed this in lines L34, L29-46, L47-49, L65-68, L73, and L139-141. We also extended the discussion on SAR backscatter over ablation areas with this addition to the manuscript at L118: **"Like in the accumulation zone, the surface melting response in the ablation zone will dominate the seasonal trends in backscatter because of absorption from liquid water at the surface over both bare-ice and debris-covered portions of ablation areas. Although the absolute fraction of backscatter at C-band frequencies over debris covered portions of ablation zones attributed to volume scatter is not well known, there is evidence that for low frequencies it can account for a majority of radar observations (Huang, et al. 2017)."**

*The method does not appear to work over ablation areas. It is unclear from the results presented if this is an isolated issue with debris-covered areas or if it is an issue with clean ice as well because the example figures were only for debris-covered glaciers. Given debris-covered areas are highly prevalent in HKH, this is a major issue. This is clearly apparent from Figure 7 Top. The Khumbu Glacier is debris-covered below the Khumbu icefall. The interpretation of the SAR signal on this debris-covered area is that it indicates refreeze onset between DOY 250-270, which is in September. This is highly inaccurate and highlights the lack of validation within the study. For example, this part of the debris-covered ice is clearly still melting in September (Rowan et al. 2020; Journal of Glaciology). Similar issues appear to persist with the Freeze Onset in Figure 8C,D.*

We find a strong seasonal loss in radar backscatter over glacier surfaces in all regions and elevation ranges of the HKH as illustrated in Figure 4B. Given our understanding of the physical radar response at these frequencies, these signals are resultant from absorption by liquid water at or near the glacier surface and are not easily explained by other physical processes. Our analysis excludes locations that do not satisfy the z-score metric (z > 2). Glacier surfaces in the ablation zone, including areas of debris cover, are included in our analysis because they exhibit a radar response indicative of surface melting and satisfy the z-score metric.

A strong reduction in surface backscatter is observed seasonally over the Khumbu Glacier, above and below the Khumbu icefall, during 2018, 2019 and 2020. These radar signals satisfy the z-score metric and we estimate the average yearly refreeze onset to be September 17the (DOY 260). Rowan et al., (2021, Journal of Glaciology) include a record of surface and below-surface temperature measurements on the Khumbu Glacier for the year 2014. It is apparent in Rowan Figure 2 (below) that mean daily air temperatures around the Khumbu

icefall (KH4) are consistently below zero in late September, 2014. This is similar to the refreeze timings especially given the 12-day repeat of the SAR observational record. Although ablation is observed to persist until October 22 at KH4, there is a disconnect between surface conditions (e.g. air temp.) and melting at the debris/ice interface. It is not surprising that melting under thick layers of debris cannot be observed directly using radar. A rigorous comparison with this data is extremely difficult since these measurements are only from one year and observations are not contemporaneous to the Sentinel-1 observational record.

[Figure]

**Fig. 2.** Daily off-glacier and on-glacier meteorological data. (a) Mean daily air temperatures measured at Khumbu Glacier, Changri Nup Glacier and the Pyramid Observatory in 2014, (b) daily precipitation amount measured at the Pyramid Observatory in 2014 (as rain plus snow in water equivalent), (c) mean daily relative humidity measured at Changri Nup Glacier in 2014 and 2016, and (d) mean daily air temperatures measured at Khumbu Glacier in 2015, Ngozumpa Glacier in 2002 and Changri Nup Glacier in 2016.

Rowan Figure 2: Air temperature record from Rowan et al. (2021) illustrating below-zero temperatures at the Khumbu icefall (KH4) starting in late September, 2014.

*Rowan, A. V., Nicholson, L. I., Quincey, D. J., Gibson, M. J., Irvine-Fynn, T. D., Watson, C. S., ... & Glasser, N. F. (2021). Seasonally stable temperature gradients through supraglacial debris in the Everest region of Nepal, Central Himalaya. Journal of Glaciology, 67(261), 170-181.*

*It's unclear if similar issues exist for the melt onset signals as well. The color bar in Figure 8A,B is too hard to read to discern if melt onset appears to be happening as early as March 1st, which would be unlikely. However, the fact that the color bar is shows DOY 60 suggests there are some areas where it is melting at this time; otherwise, why stretch the color bar outside the values shown in the figure? The issues with the freeze onset and melt onset and lack of any validation, in my opinion, undermine the entire study.*

We have corrected the extent of the color bar in Figure 8 (Figure AR3). The color bars were stretched erroneously to the maximum extent of the data across the HKH and we thank the reviewer for pointing this out. Across the three years of retrievals, the average minimum melt onset within the Central Himalaya occurred at day of year 82 between 3200m - 3300m a.s.l.. The average minimum melt onset in the Karakoram across the record occurred at day of year 93 and at the lowest elevation bin (3000m – 3100m a.s.l.). We have scaled the color bars for Figure 8 to

represent the data accordingly.

[Figure]

**Figure AR3. Melt retrievals averaged over the calendar years 2017-2019 in the Central Himalaya and Karakoram regions. (A) Mean melt onset (DOY) in the Central Himalaya. (B) Mean melt onset (DOY) over the Siachen glacier in the Karakoram region. (C) Mean melt offset (DOY) in the Central Himalaya. (D) Mean melt offset (DOY) over the Siachen glacier in the Karakoram region. Data overlay a 30m Shuttle Radar Topography Mission (SRTM) DEM hillshade (Farr, 2007).**

*One recommendation is to put the results and interpretation into better context of glacier zones. For example, the discussion/interpretation primarily focused on the liquid water in frozen snow and percolation faces, but the results that were being interpreted were aggregated from 3000-7500 m a.s.l. The lower elevations are clearly in the ablation zone, i.e., clean ice or debris-covered ice. Therefore, it's unclear how much of the discussion of the snowpack and percolation faces is warranted. Or is this partially being included to discuss seasonal snowpack covering the ablation zone? Including a rough estimate of the glacier zones, even if they are roughly estimated based on median elevation or some other metric, and their respective hypsometries may provide some useful context for interpreting the results. Otherwise, results/discussion like L406-414 come across as interpreting the entire glacier as being in the percolation zone.*

We fully agree with the reviewer, it would be ideal to discuss results in context of glacier zones. However, we are not aware of a detailed mapping of glacier zones at a scale appropriate for data record. We use aggregates of elevations bins at 1000 m intervals to communicate our results in Table 3.

Line 406 refers to results over specific elevation bins where delayed refreeze is apparent (i.e. at higher elevations). We have changed the sentences beginning with L406 to clarify that this paragraph is meant specifically to discuss observations of delayed refreeze at high elevations and how these observations relate to meltwater retention specifically within percolation zones: **"Signals of delayed refreeze are observed at elevation ranges similar to greatest z-score across summary statistics of FO (*Supplementary Fig. S2*). Notably, we find specific high elevation ranges in select catchments in the western sub-regions (Eastern Hindu Kush, Western Pamir, and Karakoram) and some eastern sub-regions (Tanggula Shan, Nyainqentanglha, Eastern Tibetan Mountains, and Hengduan Shan) where there is a signal of delayed refreeze apparent in summary statistics. Although sub-regional aggregate FO statistics do not show delayed refreeze in larger sub-regions (i.e. the Central Himalaya), we observe signals of delayed refreeze on individual glaciers within the Central Himalaya indicative of meltwater retention within percolation facies (Figure 7)."**

*Given the major issues with the ablation area, I would suggest either (i) limiting the study to only accumulation areas, or (ii) limiting the study to only clean-ice glaciers. Either way, there needs to be significantly more validation performed to provide confidence in the results. This validation should ideally be done for both the ablation and accumulation areas, albeit that the accumulation zones interpretation should be much stronger as they are based in theory as the authors clearly discuss in the main text and mention in response to a previous reviewer.*

There is a well-known shortage of validation data (e.g. automated weather stations) throughout the HKH. Therefore, it is vital that we advance alternate means of glacier monitoring like remote sensing. We contend that the sensitivities of radar to liquid water at the glacier surface make it an ideal instrument to record glacier surface melting. The physical response of SAR to liquid surface-water is well established in various applications. We fully agree, uncertainty assessments would be ideal if validation data was available. To our knowledge, there is no remote sensing available at the spatial and temporal resolution required to define this uncertainty. For additional discussion on this topic, please see Author Responses (AR) to Reviewer #1 during the first round of reviews. Specifically, AR3, AR9, AR14, available at this link.

*This validation is important as it is unclear how sensitive the results are to various aspects of the methods. For example, is there any sensitivity to the chosen dB threshold of 3? A previous reviewer asked for a sensitivity analysis, but the author's simply responded that this was*

*conservative based on literature values. Even if this is only done for a handful of glaciers, a sensitivity analysis would provide more confidence. Similarly, what about using ascending vs. descending orbits? The choice of orbit direction clearly affects the interpretation of the signal at high elevations (Figure 5) and yet for the full region a composite was used. If this composite is used moving forward, it'd be good to see some sensitivity/error analyses because it appears to have a strong impact on the results, but perhaps this is only for high altitudes.*

We use a universal threshold (i.e. not spatially or temporally varying) that is well established in prior studies and easy to implement on a large-scale dataset. As indicated in the text, this value was first suggested in Ashcraft and Long (2001) using a simple scattering model and is equivalent to a loss of one-half power. A spatially or temporally varying threshold requires a more complex approach but given the computational requirements and large-scale dataset (i.e. >100 Terrabytes) we believe this is beyond the scope of our current study.

*Ashcraft, Ivan S., and David G. Long. "Azimuth variation in microwave backscatter over the Greenland ice sheet." IGARSS 2001. Scanning the Present and Resolving the Future. Proceedings. IEEE 2001 International Geoscience and Remote Sensing Symposium (Cat. No. 01CH37217). Vol. 4. IEEE, 2001.*

*Similarly, were the high-altitude automatic weather stations used for validation? The energy balance modeling likely has its own issues (e.g., sublimation could be important here?), but the modeling appears to be used to validate the SAR signal (L320). If it is used for validation, does that mean that the SAR signal overestimates the amount of melt by 33-43% (L345)? This would appear to be a considerable amount of error.*

We fully agree with the reviewer's assertion that surface energy balance modeling at high elevations is complex and therefore requires considerations that are beyond the scope of this exercise. Surface energy balance outputs are used to confirm that widespread signals observed, for the first time, in SAR time series are very likely due to surface melting. We have no direct measurements of liquid water in snow and firn for full validation. Agreement between energy balance models and radar demonstrate that surface melting at high elevations is difficult to predict using only measurements of air temperature. Since the radar response to liquid water is largely unambiguous there is not a strong argument to attribute error from this modeling exercise to SAR, especially where modeling exercises are not well constrained (e.g. sublimation).

*Specific Comments*
*Given there's no word limit, I would suggest writing out acronyms like melt onset and freeze onset to make the study more readable to the average reader.*

We have limited the acronyms that we define in the paper to freeze onset (FO) and melt onset (MO). These acronyms signal to the reader that we are referring to our record of glacier surface melting. This limited use of acronyms should be accessible to the average reader, we are careful not to add any additional acronyms to the text.

*L34: "Ice caps" have a particular meaning. In HKH, they are primarily (if not all) mountain glaciers or valley glaciers.*

We have removed references to "ice caps" in L34 and instead referred to "mountain glaciers."

*L29-46: This introduction bounces back and forth between discussing global issues (e.g., contributing 25% of sea level rise) to HKH specific issues (e.g., freshwater for 2 billion people). It's useful to show how HKH changes fit into the larger picture, but I'd be conscious of this change in scales to make it easier to read.*

The implications of glacier wasting in the HKH present both regional and global hazards. Stylistically, we sought to illustrate a broad overview of the hazards associated with glacier wasting in the HKH as those hazards range in scale from local (i.e. outburst flooding) to global (i.e. sea level rise).

*L47-49: Litt et al. (2019) focuses on two glaciers over a period of a several years. It does not address "projected disturbances". More appropriate reference would be one of the GlacierMIP studies (e.g., Hock et al. 2019 (Journal of Glaciology) or Marzeion et al. 2020 (Earth's Future)). The same is true for L55, which discusses projections but does not mention relevant studies where these uncertainties exist.*

We have changed the sentence beginning at L48 accordingly: **"Substantial uncertainties exist in projected disturbances associated with a changing climate, environment, and hydrologic regime across the greater Himalayas due in part to a lack of observations of *in situ* hydrology and meteorology at high elevations (Hock et al., 2019; Litt et al., 2019; Marzeion et al., 2020)."**

We have changed L55 accordingly: **"Although the general trajectory of changes to the HKH cryosphere is understood (i.e. accelerated glacier mass loss on a decadal scale in the Central and Eastern Himalaya) (Fujita and Nuimura, 2011), a consensus in projecting changes to HKH hydrology is lacking largely because of missing *in situ* snow and ice monitoring data across these glaciated river basins (Marzeion et al., 2020)."**

*L65-68: Statement concerns inability of studies to capture variability in patterns and magnitude of melting, but only references a study that measures the geodetic mass change (Brun et al. 2017). The more recent study by Shean et al. (2020; Frontiers in Earth Science) would be appropriate to reference here as it provides refined estimates of mass change compared to Brun et al. (2017). Furthermore, the recent study on projections in this region (Rounce et al. 2020; Frontiers in Earth Science) explicitly captures the variability reported by these measurements using a degree-day model.*

A degree-day model would not capture the melt occurring for several months of the year at elevation ranges above the 0ºC summer isotherm, as presented in this study. Brun, et al. (2017) is cited to highlight sub-regional variability in the sentence beginning L66

*L73: "operational monitor" – consider changing "operational monitoring system" perhaps?*

We have changed this sentence accordingly in addition to all references to "operational monitor" made throughout the text at L23, L73, and L569.

*L127 – this would be useful for assessing all models of glacier ablation, not just energy balance models.*

We have changed the sentence accordingly: **"It is possible that intense incident solar radiation is driving these melt processes at elevations above the 0ºC summer isotherm (Matthews et al., 2019) across the entirety of the HKH, and that the sensitivity of SAR backscatter to changes in the glacier surface melt/freeze condition as seen when water transitions between solid and liquid phases provides a real alternative to temperature elevation lapse rate estimates of melting (Litt et al., 2019) for assessing models of glacier ablation."**

*L137 – for The Cryosphere I would suggest using more standard glacier mass change terms (e.g., mass loss, mass balance, mass change) as opposed to the term "wasting" that is less frequently used.*

We have modified language around glacier mass loss accordingly by changing references to "mass wasting" to "mass loss" at L29, L38, L532, L533, L537, and L539.

*L139-141 – appears to imply that Karakoram, Kunlun Shan, etc. are not affected by increases in global average temperature. I'd suggest references Karakoram anomaly and recent studies finding that they are starting to lose mass (e.g., Farinotti et al. 2020 (Nature Geoscience)).*

This sentence was meant to highlight sub-regional variability in melt patterns. We have changed this sentence accordingly: **"Glacier wasting in the HKH is heterogeneous and the increase in global average temperature has caused wasting of mountain glaciers across all HKH sub-regions (Farinotti et al., 2020; Gardelle et al., 2012)."**

We have also changed the sentence beginning at L535 accordingly: **"Sub-regions with slower mass loss (Eastern Hindu Kush, Western Pamir, Karakoram, Tibetan Interior) show on average one month less of melt duration relative to regions with accelerated mass loss."**

*L148 – is there a reference for this ICIMOD dataset?*

We have added a citation for this shapefile at L148.

*L158-171 – this is a lot of detail on the glacier outlines used in a study that I don't see being relevant. If debris-covered areas are an issue, there's no mention of datasets that explicitly delineate it (Scherler et al. 2018 (Geophyiscal Research Letters); Herreid and Pellicciotti 2020 (Nature Geoscience)). This should be discussed.*

We thank the reviewer for this suggstion. We agree that too much discussion of the process for producing glacier outlines is not appropriate for this paper. We mention the issue of debris covered glaciers and the production of glacier outlines in a single sentence and as an aside (L166-169). We mention the use of SAR interferometry to delineate debris-covered glaciers.

*L189 – "this" should be "the" or "the timeframe of this study"*

We have incorporated this suggestion into the text at L189.

L204 – suggest stating the name of the cloud-computing platform here, i.e., Google Earth Engine, to make this explicit for readers as opposed to only mentioning it at the end of the paragraph.

We have restructured L204: **"A cloud-computing platform and application programming interface (Google Earth Engine) with pre-processed radiometrically terrain corrected Sentinel-1 A/B data was used to detect melt characteristics across the region (Gorelick et al., 2017)."**

*L216 – It is my understanding that these are both on Khumbu Glacier. Therefore, this should be "over a high elevation glacier", not glaciers.*

We have restructured L216 accordingly: **"Measurements from two automated weather stations (AWS) are used to estimate surface energy balance (SEB) and evaluate surface melting conditions over the Khumbu Glacier."**

*L245 – Why is this weather station not stated in Section 2.4?*

We thank the reviewer for pointing this out and are happy to have made the following changes to better organize presentation of AWS data in the paper. We have moved Table 2 to Section 2.4 while also adding the Everest AWS data to Table 2.

We have rewritten the sentence at L215 as follows: **"Measurements from two automated weather stations (AWS) are used to estimate surface energy balance (SEB) and evaluate surface melting conditions over the Khumbu Glacier and measurements from two additional AWS are used to calculate temperature-elevation lapse rates for comparison with melt retrievals (Table 1)"**

We have appended the following sentence at L222: **"AWS data collected within the Langtang Valley are used to estimate temperature elevation lapse rates following prior studies and serve as data for comparison with Sentinel-1 backscatter values (Table 1) (Shea, 2016). "**

*L247 – Introduction stated that assuming 0 degC threshold is poor assumption and was part of the motivation for this study. May want to preface that this generally works well, except for at*

*extreme altitudes where other processes (e.g., sublimation) may be important (if that is the point that is trying to be stated)?*

We thank this reviewer for this suggestion and strongly agree that temperature elevation lapse rates do well to capture the processes of glacier melt within the HKH at elevations where the maximum daily average temperatures exceed 0ºC, as stated in L567. We however contend that our statements regarding temperature-elevation lapse in fact working well at elevations where mean daily air temperatures exceed 0ºC (e.g. through agreement presented in L247, statements in L494, L567) will suffice.

*L296 – It's unclear how debris-covered areas were identified (see previous comment). Nonetheless, not including debris meant between 30-48% of the ice in the ablation area is not being monitored (Kraaijenbrink et al. 2017; Nature). It would be good to state how much over the actual glacier area was able to be monitored, and if statements are being made about the ablation area (like that being mentioned here), then stating the percentage of the area in the ablation area being monitored is relevant as well.*

In Table 3, we tabulate the area over which melt signals were retrieved within each sub-region following median window filtering. In Figure 4, Z-score data by elevation shows that there are only two sub-regions (Eastern Tibetan Mountains and Eastern Hindu Kush) where the mean z-score is well below the threshold of 2, suggesting that these low elevation areas (~3000m - 34000m a.s.l.) were the only portions of the study region where Sentinel-1 did not record a strong enough signal related to seasonal melt processes.

We have restructured L309 in order to articulate more clearly that there are two elevation ranges and regions where the z-score test show an inverse response to seasonal melt: **"Mean seasonal melt magnitude averaged over 100m elevation bins over all three calendar years of data shows strong (z > 2) melt signals across glacio-climatic sub-regions and across all elevation ranges of significant glaciation except below ~3,400m a.s.l. in the Eastern Tibetan Mountains and Eastern Hindu Kush sub-regions."**

*L307 – if the debris-covered area is excluded, then how are statements being made concerning those areas? Is the assumption at the regional scale that when evaluating variability, the debris-free areas that were measured at lower elevations are representative of the debris- covered areas? If so, this should be stated explicitly, as it's confusing within its present form. Note that the Khumbu Glacier example clearly shows debris-covered areas being included.*

Exclusion within any area due to the presence of liquid water not dominating the signal will be conducted on a per-pixel basis and interpolated over at the end of product generation between areas where a melt signal was retrieved.  Areas are excluded that do not show a seasonal sensitivity to the backscatter signal representative of the presence of liquid water at or near the surface.

*L322 – "are" should be "is"*

There are multiple weather stations installed at the Khumbu glacier.

*L324 – If the energy balance modeling is an important part of this study, which it appears to be, then additional detail should be provided in the main text. At a minimum, this should include the values used to force the model (i.e., those in Table SI) as well as relevant information pertaining to the timestep and the terms considered (i.e., Equation 1 in the supplement).*

We contend that inclusion of model parameters is not central to this paper as the model was parameterized identical to Matthews, et al. (2019). All relevant model parameters are included in the supplementary section.

*L330-350 – these are results, not methods.*

This is a confirmation that our methodology is correctly interpreting the melting signal. This information is not included in the resulting data record that we derive.

*L337-339 – I was thinking this same point, so I'm glad the authors brought this up. However, the timing of satellite overpasses is known (Figure 5 caption), so why was this not performed? This appears to be the primary validation of how well SAR performs, and one of the primary conclusions of the study that SAR can detect melting where models otherwise wouldn't, so it should be presented in a rigorous analysis to provide confidence.*

The SEB model was run at half temporal resolution at two hour time steps. When matching the overpass timing of the descending and ascending passes (6:00am and 6:00pm respectively) we find that the SEB determined melting duration is largely the same and does not change the interpretation of our data. In order to avoid this notable confusion in the text, we thank the reviewer for pointing this out and have removed the following sentence at L337: **"A more robust comparison would match the timings of satellite overpasses and meteorological observations and acknowledge that some disagreement between melting estimates is resultant from this difference."**

*L351-355 – High degree of confidence from two weather stations, where the days of melting appears to be overestimated by 33-43% (L345) seems to oversell the results. I agree that this is a challenging topic, especially since there are uncertainties with the SAR data and uncertainties with the energy balance model (unfortunately, there's no validation data for the energy balance model, so are these differences due to the SAR data or the energy balance model – likely both?).*

We have a high degree of confidence melt is occurring where SEB results and Sentinel-1 retrievals show that the glacier surface is melting despite air temperatures below 0ºC.

*L355 – "HMA" (High Mountain Asia) is not defined in text. It's used later in text as well. I'd suggest using HKH throughout.*

The "HMA" abbreviation was mistakenly included. We have changed this to HKH at lines L355,

L531, and L533.

L386 – except for debris-covered areas, no (L297)?

There are only two regions where the mean z-score is less than 2, which we highlighted in the response to the specific comment on L296.

*Table 3 – caption and table are opposite directions making it very difficult to read. Also, 1 km elevation bins are very coarse considering that some regions (if not most regions) have most of their glacier area spanning 1-2 km (Figure 4 Bottom). This means that the statistics shown for the other elevation bins is for a tiny fraction of the glacier area, which in my opinion detracts from the overall value of this table. The 100 m bins in Figure 4 are much more meaningful.*

The data presented in Figure 4 and the Supplementary Figures on sub-regional melt timing and z-score sufficiently summarize the data for the purpose of this communication. Those interested in working with the data will be able to retrieve it and calculate statistics relevant to their study regions at 90m maximum spatial resolution.

*Figure 6 – this figure is difficult to read/interpret due to the lines. I would suggest changing both the circles and the squares to lines. The circles and squares are clearly separated, so you could simply add a note that the melt onset is on the left and freeze onset on the right, which is intuitive anyways. This would enable the reader to follow the lines and determine how the trends vary. It will also make it easier to see the 12 subregions, which currently are on top of one another and hard to discern any information from. This will also make it easier to see statements like L400 where it does not follow a linear trend, when the current form of Figure 6 looks like roughly speaking higher elevations are around DOY 250-270 and lower elevations are around 280-330, so while the linear trend is not as strong, there still appears to be a trend.*

We have reformatted the figure to include lines instead of shapes (Figure AR4) and inserted it into the manuscript at L415. For more detailed illustrations on non-linear trends in melt patterns, please see Supplementary Figure 2.

[Figure]

**Figure AR4. Mean melt onset (MO) and freeze onset (FO) summarized in 100m elevation bins using the 30m SRTM digital elevation model (Farr, 2007) and 12 HiMAP sub-regions (Shean, 2020). The blue to red color scale indicates the longitude of the HiMAP region centroid, where the westernmost regions are shown in dark blue and eastern most shown in dark red.**

*L393-396 – is this supported by the energy balance modeling?*

SEB models are at point-locations and therefore will not capture melt/freeze characteristics as they vary across elevations. Additionally, C-band SAR will be sensitive to the presence of liquid water across a depth on the order of meters in the percolation zone, which may not be captured in SEB models.

*L429-433 – Khumbu Glacier is highly debris-covered below the Khumbu icefall, which the authors appear to interpret as a signal of refreeze onset. This is highly inaccurate. Figure 7 (Top) suggests the Khumbu Glacier would be refreezing on DOY 250-270 (sometime in September) when the debris-covered ice is clearly still melting (Rowan et al. 2020; Journal of Glaciology). Figure 8C,D show similar issues. See major comment.*

Please see response to the major comment above.

*L533 – Shean et al. (2019a) does not exist. Unclear therefore where the trends for 2000-2010 are coming from and why trends in later decades covering the time period observed (i.e., 2000- 2018 from Shean et al. 2020) are not used.*

We have corrected this reference to refer to Shean, et al. (2020) at L533.

*L557 – surface energy balance models "are constrained" by SAR data indicates that the SAR data is used to calibrate the energy balance models. This is in direct contrast to earlier where the energy balance models were being used as validation of the SAR observations (L320).*

We have changed the sentence beginning at L557 accordingly: **"Melt conditions in surface energy balance models of glacier melt driven by *in situ* meteorological data from Mount Everest fall within the date ranges of melt retrievals recorded in Sentinel-1 SAR data."**

---

## Author Response (AR4)

Dear Melody,

Many thanks for your comments and corrections. We have addressed the following:

*Line 16: (83,102 km2 glacierized area), defined using optical observations may read better as (83,102 km2 glacierized area, defined using optical observations) [optional]*

We have corrected this in the current manuscript.

*Line 111: Importantly for melt retirevals -> Importantly for melt retrievals*

We have corrected this in the current manuscript.

*Line 116: thus progressivly -> thus progressively*

We have corrected this in the current manuscript.

*Line 313: data is -> data are*

We have corrected this in the current manuscript.
*Figure 6 caption: Mean melt onset (MO) and freeze onset (FO) -> Mean melt onset (MO, left) and freeze onset (FO, right),*

We have corrected this in the current manuscript.

*Figure 8 caption: (C) Mean melt offset (DOY) in the Central Himalaya. (D) Mean melt offset -> (C) Mean freeze onset (DOY) in the Central Himalaya. (D) Mean freeze onset*
We have corrected this in the current manuscript.

*Supplementary material, line 15: The $QSW$ -> $QSW$*

We have corrected this in the current supplement.

*Please add the link to the code and data in the Code and Data Availability section.*

We have added a DOI link to the data repository at NSIDC and the code repository on github.

With best wishes,

Nick